# Acute stress enhances adult rat hippocampal neurogenesis and activation of newborn neurons via secreted astrocytic FGF2

Elizabeth D Kirby[1†], Sandra E Muroy[2], Wayne G Sun[2], David Covarrubias[2], Megan J Leong[1], Laurel A Barchas[3], Daniela Kaufer[1,3]*

[1]Helen Wills Neuroscience Institute, University of California, Berkeley, Berkeley, United States; [2]Department of Molecular and Cell Biology, University of California, Berkeley, Berkeley, United States; [3]Department of Integrative Biology, University of California, Berkeley, Berkeley, United States

**Abstract** Stress is a potent modulator of the mammalian brain. The highly conserved stress hormone response influences many brain regions, particularly the hippocampus, a region important for memory function. The effect of acute stress on the unique population of adult neural stem/progenitor cells (NPCs) that resides in the adult hippocampus is unclear. We found that acute stress increased hippocampal cell proliferation and astrocytic fibroblast growth factor 2 (FGF2) expression. The effect of acute stress occurred independent of basolateral amygdala neural input and was mimicked by treating isolated NPCs with conditioned media from corticosterone-treated primary astrocytes. Neutralization of FGF2 revealed that astrocyte-secreted FGF2 mediated stress-hormone-induced NPC proliferation. 2 weeks, but not 2 days, after acute stress, rats also showed enhanced fear extinction memory coincident with enhanced activation of newborn neurons. Our findings suggest a beneficial role for brief stress on the hippocampus and improve understanding of the adaptive capacity of the brain.

*For correspondence: danielak@berkeley.edu

[†]Present address: Neurology, Stanford University, Stanford, United States

**Competing interests:** The authors declare that no competing interests exist.

## Introduction

Stress is a powerful and essential mediator of mammalian behavior. Proper response to a perceived stressor facilitates survival at the individual level and species propagation at the population level. Despite this necessity, stress responses can become maladaptive. Chronic stress, for example, leads to a host of adverse health consequences, including cardiovascular disease, obesity, depression, and exacerbation of neurodegeneration (*McEwen, 2004*). Acute stress, defined as a single exposure on the scale of minutes to hours without cycles of recovery and re-exposure, has proven more enigmatic.

One model of stress effects on the brain, an inverted U function, explains the variable consequences of acute stress for brain health (*Lupien and McEwen, 1997*). In this model, while severe or prolonged stressors are detrimental, brief or moderate stressors actually enhance neural function. Behavioral studies focusing on the memory functions of the hippocampus have demonstrated such a relationship in rodents, where moderate stress enhances memory performance yet more severe stress impairs it (*Conrad et al., 1999*).

The hippocampus is exquisitely sensitive to stress and the primary stress hormone class, glucocorticoids (GCs). Within the dentate gyrus (DG) sub-region, in particular, there exists a high density of GC receptors that respond to elevated circulating GCs (*De Kloet et al., 1998*). In addition, the DG is strongly connected via the entorhinal cortex and medial septum to the basolateral amygdala (BLA), a

**eLife digest** A little stress can be good for you. Just over 100 years ago, psychologists Robert Yerkes and John Dodson suggested that cognitive performance improves as stress increases, although it falls off again if stress levels continue to rise. The hippocampus is a key brain region for both memory and the regulation of emotion, and is highly sensitive to the main class of stress hormones, glucocorticoids. One particular subregion of the hippocampus, the dentate gyrus, contains a high density of glucocorticoid receptors, and is also notable for being one of only two regions in the adult mammalian brain that can give rise to new neurons via a process called neurogenesis.

Chronic stress is known to impair memory and to reduce neurogenesis. However, the effects of acute stress are less clear-cut: early studies suggested that it suppressed the generation of new neurons, whereas several recent studies have observed no effect. Other work has shown that acute stress increases the expression of growth factors—substances that stimulate cellular growth and proliferation—which would seem to suggest that stress could enhance neurogenesis.

Now Kirby et al. have obtained further insights into the effects of acute stress on the proliferation of cells in the dentate gyrus. Exposing rats to a moderate acute stressor, namely being restrained for a few hours, led to increased neurogenesis in the dorsal, but not ventral, hippocampus. Injecting rats with the stress hormone corticosterone had the same effect. In both cases, the enhanced neurogenesis was accompanied by increased expression of a growth factor called FGF2, which is produced by glial cells called astrocytes.

Intriguingly, Kirby and co-workers found that the stressed rats performed better than control animals in a memory test. Moreover, the beneficial effects were seen if the rats performed the task 2 weeks after their stressful experience, but not if they performed the task 2 days after being stressed. This is pertinent because new neurons in the dentate gyrus become functional 2 weeks after being generated, which suggests that the stress-induced increase in neurogenesis could account for the rats' improved memory.

The work of Kirby and co-workers has thus identified a mechanism by which moderate acute stress could have beneficial effects on cognition. Given that acute stress can be harmful in other instances—leading, for example, to post-traumatic stress disorder—further work is required to identify the factors that determine whether a response to stress is adaptive or pathological.

brain region involved in emotional processing and an important mediator of many stress effects on the hippocampus (*McGaugh, 2004*). Both of these mediators of stress (GCs and BLA input) have been shown to regulate the unique population of neural progenitor cells (NPCs) that reside in the adult DG (*Kirby and Kaufer, 2009*; *Kirby et al., 2012b*).

Dentate NPCs proliferate and give rise to new neurons throughout the lifespan in several mammalian species, including rats, mice and primates (*Abrous et al., 2005*; *Kirby and Kaufer, 2009*). They become electrophysiologically active, integrate into local circuitry, play important modulatory roles in hippocampal memory function (*Abrous et al., 2005*) and can respond to stress and stress hormones at multiple phases of development (*Kirby and Kaufer, 2009*). Moreover, recent work indicates that newborn, immature neurons integrate multiple signals more readily than mature neurons and that they have enhanced excitability, possibly contributing to a disproportionately large role in new memory formation (*Marín-Burgin et al., 2012*).

Numerous studies show that chronic stress, in addition to impairing memory function, suppresses proliferation, survival and differentiation of new neurons in the adult DG (*Wong and Herbert, 2004*; *Mirescu and Gould, 2006*; *Kirby and Kaufer, 2009*). The effect of acute stress on neurogenesis, however, is unclear. While early work indicated suppression of proliferation following acute stress or GC injection (*Cameron and Gould, 1994*; *Gould et al., 1997*), subsequent studies have yielded mixed results, often reporting no change in proliferation following a variety of acute stressors (*Thomas et al., 2006*; *Thomas et al., 2007*; *Dagyte et al., 2009*; *Hanson et al., 2011*). In contrast, investigations of hippocampal growth factor secretion have shown that acute stress enhances expression of mitogenic growth factors such as basic fibroblast growth factor (FGF2) and nerve growth factor (NGF) (*Mocchetti et al., 1996*; *Molteni et al., 2001*), implying a potential for increased neurogenesis following acute stress. Indeed, a number of interventions that stimulate GC release such as acute exercise (*Kronenberg*

*et al., 2006*) and acute sexual experience (*Leuner et al., 2010*) actually increase neurogenesis in the adult hippocampus. Combined, these studies suggest that adult hippocampal neurogenesis may follow an inverted U function similar to hippocampal memory—decreasing following chronic stress yet increasing in response to acute stressors.

We examined the effect of several forms of acute stress on adult hippocampal neurogenesis, seeking to resolve the contradictory evidence for enhanced vs impaired hippocampal plasticity. We found that acute stress or corticosterone (CORT, the primary rat GC) administration increased dorsal but not ventral hippocampus cell proliferation in adult rats. This increase was not dependent on input from the BLA and was accompanied by an increase in FGF2 expression in dorsal hippocampal astrocytes. Furthermore, we show that astrocyte-secreted FGF2 is necessary for CORT-induced enhancement in NPC proliferation in vitro. 2 weeks after acute stress, when newborn neurons are first becoming functional, we also find enhancement in hippocampus-dependent memory accompanied by enhanced activation of newborn neurons. These findings have important implications for understanding regulation of hippocampal plasticity in the face of environmental challenge and in distinguishing adaptive vs pathological stress responses. Moreover, they suggest that stress effects on adult neurogenesis may follow an inverted U function similar to that already demonstrated for hippocampal memory function (*Lupien and McEwen, 1997*).

## Results

### Acute stress increases dorsal hippocampus NPC proliferation

To investigate the effect of acute stress on adult neurogenesis in the DG (*Figure 1*), we chose three common models of acute stress in rodents: 30 min novel environment, 30 min footshock or 3 hr immobilization. Rats were handled for 5 days prior to stress exposure then perfused 3 hr after the beginning of the stressor (*Figure 2A*). Immobilization stress significantly increased the number of cells immuno-positive for the proliferation marker Ki67 in the dorsal DG 3.23-fold above control (27.51 ± 2.74 control vs 88.99 ± 12.49 immob) while novel environment or footshock did not significantly alter Ki67+ count compared to control (*Figure 2B,C*). Plasma CORT was significantly elevated above control levels both 30 min and 3 hr after stressor initiation in immobilized rats but not in novel environment- or footshock-exposed rats (*Figure 2D*). This finding suggested that an increase in CORT might underlie the immobilization-induced increase in DG proliferation. To test that hypothesis, we next habituated handled rats to daily oil injections for 3 days, injected them with exogenous CORT (0, 5 or 40 mg/kg body weight) and assessed cell proliferation 3 hr later (*Figure 2A*). 40 mg/kg CORT significantly increased the number of Ki67+ proliferative cells in the dorsal DG 1.92-fold above oil-injected controls (26.19 ± 2.83 0 mg/kg CORT vs 50.29 ± 7.74 40 mg/kg CORT; *Figure 2C,E*). Plasma CORT levels were also consistently elevated 30 min and 3 hr after injection (*Figure 2F*), similar to the levels seen in immobilized rats. Notably, injection of 5 mg/kg CORT yielded similar plasma CORT levels to footshock (approximately 119 and 121 ng/ml, respectively) and also did not produce an increase in the numbers of Ki67+ cells. If rats were not habituated to injection, no difference in cell proliferation was found (*Figure 2J,K*). We next assessed short-term survival of newly-born cells following acute stress. 3 hr after the start of immobilization or CORT injection, rats were injected with the proliferative marker 5-bromodeoxyuridine (BrdU) then perfused 24 hr later. Both immobilization (*Figure 2G,I*) and 40 mg/kg CORT injection (*Figure 2H,I*) significantly increased the number of cells immunopositive for BrdU surviving 24 hr after termination of the stressor in the dorsal DG by 1.92 and 1.48-fold, respectively (34.88 ± 8.70 control vs 68.96 ± 10.23 immob; 39.80 ± 6.89 0 mg/kg CORT vs 58.95 mg/kg CORT).

### Acute stress does not increase ventral hippocampus NPC proliferation

Recent work suggests different functional roles for the dorsal and ventral hippocampus in mediating spatial memory vs emotion regulation, respectively (*Fanselow and Dong, 2010*). Adult neurogenesis may also be differentially regulated in dorsal vs ventral hippocampus; while environmental enrichment enhances neurogenesis in both dorsal and ventral DG, chronic mild stress preferentially suppresses proliferation in the ventral subregions (*Tanti et al., 2012*). In our acute stress models, we found no effect of novel environment, footshock, immobilization or CORT injection (*Figure 3A–C*) on proliferative Ki67+ cell number in the ventral hippocampus (*Figure 1B*).

### BLA input does not modulate acute stress-induced proliferation

Work by McGaugh and colleagues shows that acute stress-induced alterations in hippocampal memory and long-term potentiation depend on input from the BLA (*Quirarte et al., 1997*; *Roozendaal et al.,*

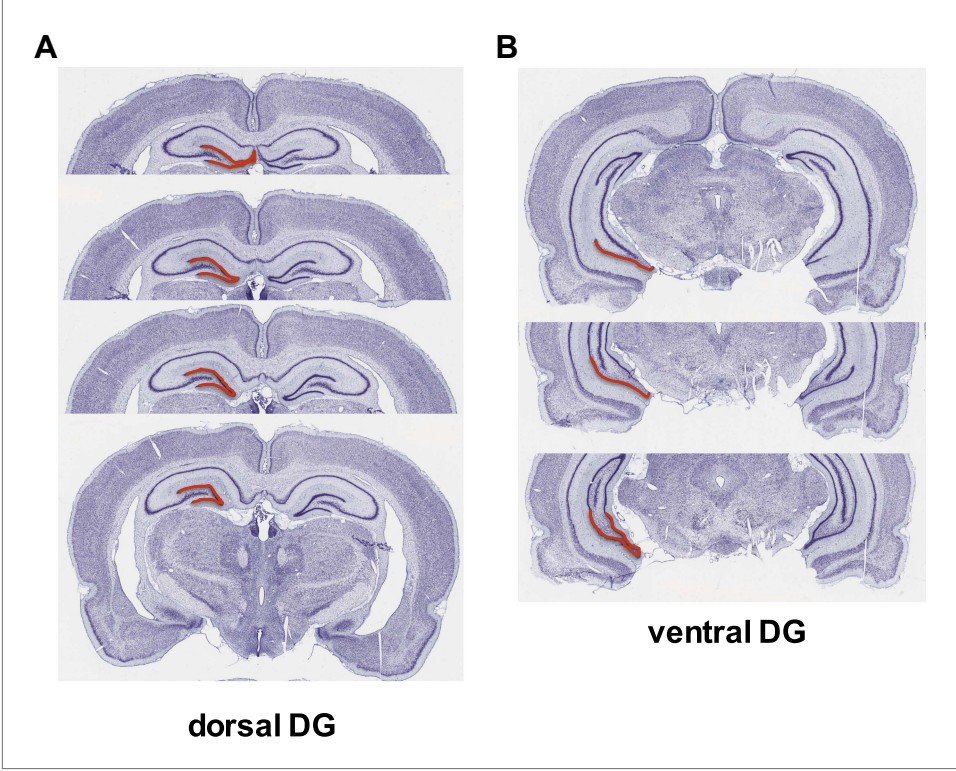

**Figure 1.** Dorsal versus ventral DG. Areas of dorsal (**A**) and ventral (**B**) dentate gyrus used for cell proliferation quantification are highlighted in red. Images are adapted from brainatlas.org.

*1999*; *Roozendaal et al., 2009b*), a fear and emotion processing center. In addition, we have previously shown that BLA input regulates adult DG neurogenesis under basal conditions and is required for the activation of newborn neurons in a fear conditioning paradigm (*Kirby et al., 2012b*). To test whether BLA input is necessary for the acute stress-induced increase in proliferation, we performed unilateral excitotoxic lesions of BLA in adult rats (as described in [*Kirby et al., 2012a*, *2012b*]). Animals were then exposed to immobilization stress or no stress control (*Figure 4A*). At the end of the stressor, each rat received a BrdU injection and was perfused 2 hr later. Plasma CORT response to immobilization was similar in lesioned and sham-operated rats (*Figure 4B*), suggesting intact hormonal stress response in BLA-lesioned, immobilized rats. Consistent with our findings in intact rats, sham-operated immobilized rats had a significant 2.6-fold increase in BrdU+ proliferative cells over no stress controls (6.37 ± 1.14 control vs 16.69 ± 2.11 immob; *Figure 4C*). BLA lesion suppressed proliferation ipsilateral to the lesion by approximately 1.5-fold (1.66-fold control, 1.44-fold immob), as we have previously reported, but did not block the stress-induced increase in proliferation (*Figure 4D*). These findings suggest that input from the BLA is not necessary for acute stress effects on adult hippocampal cell proliferation.

## CORT-induced increase in isolated NPC proliferation is dependent on astrocyte signaling

Given that the stress-induced increase in proliferation was not dependent on BLA input, we next tested whether stress-induced proliferation is a cell autonomous phenomenon. This was accomplished by quantifying the response of isolated adult rat hippocampal NPCs grown in vitro (as described in *Gage, 2000*) to acute CORT exposure. NPCs were FGF2 deprived for 24 hr then treated with 1 µM CORT (equivalent to approximately 350 ng/ml, the level measured in plasma of immobilized rats) or EtOH vehicle in either high (20 ng/ml) or low (0 ng/ml) FGF2. After 3 hr, BrdU was added to the cells and they were fixed 2 hr later (*Figure 5A*). Cultured rat hippocampal NPCs depend on FGF2 signaling for proliferation (*Chipperfield et al., 2002*). We found that while high FGF2 increased the percentage of proliferating BrdU+ NPCs over low FGF2 (1.21-fold in EtOH, 1.48-fold in 1 µM CORT), exposure to

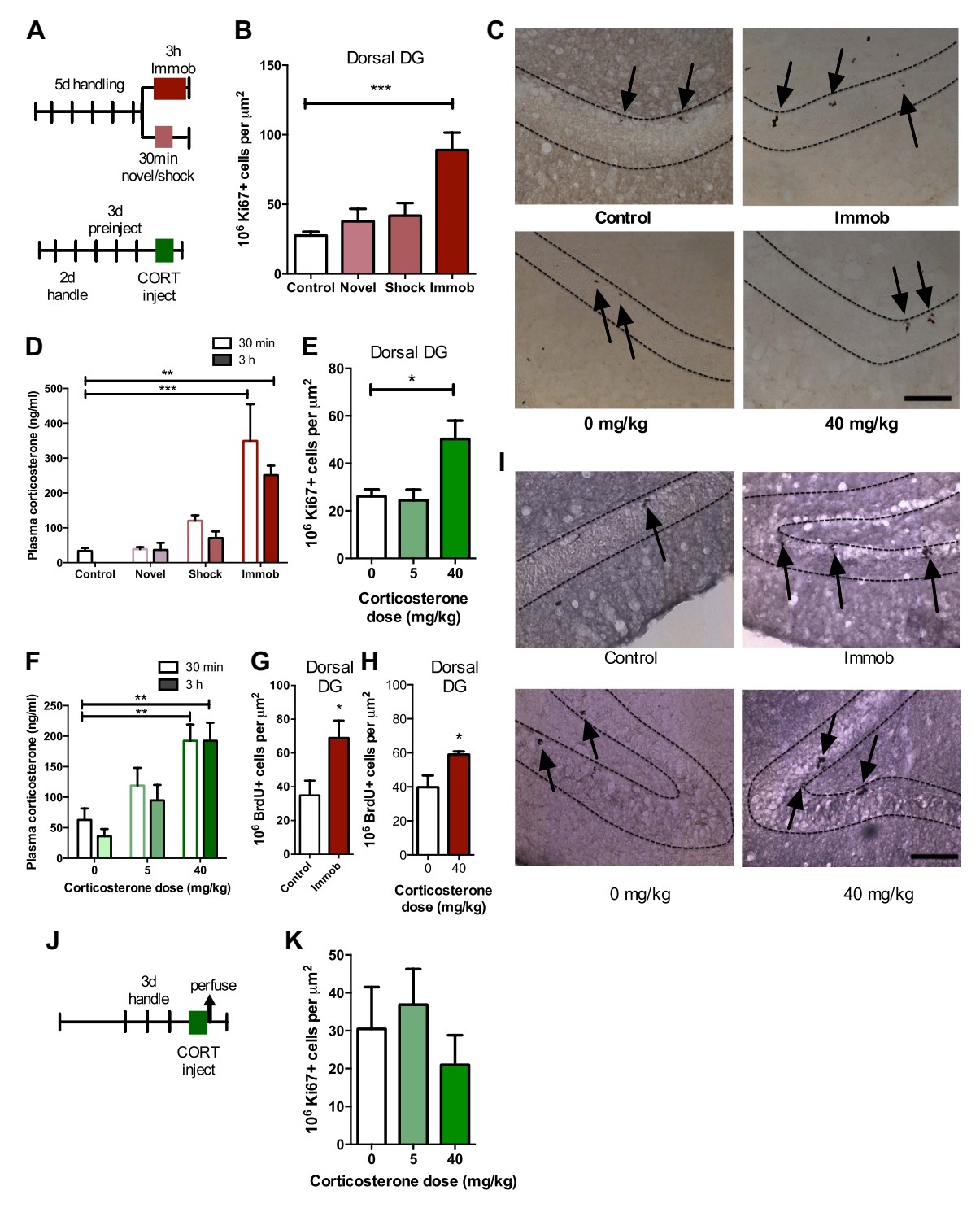

**Figure 2.** Acute stress increases adult cell proliferation in dorsal hippocampus. (**A**) Experimental timeline. (**B**) Acute immobilization increased Ki67+ cell count in the adult dorsal DG while exposure to novel environment or footshock did not significantly change Ki67+ cell count. One-way ANOVA, p<0.0001; ***q = 5.975, p<0.0001. (**C**) Representative images of Ki67+ cells (black arrows) in the dorsal DG (dashed outline) of control, immobilized, 0 mg/kg and 40 mg/kg CORT-injected rats. (**D**) Acute immobilization increased plasma CORT levels 30 min and 3 hr after the stressor began. CORT elevations caused by novel environment and footshock were not significant. One-way ANOVA, p<0.0001; ***q = 5.56, p<0.0001; **q = 4.02, p<0.001. (**E**) Acute injection of 40 mg/kg CORT increased Ki67+ cell count in the adult dorsal DG compared to 0 mg/kg oil control while 5 mg/kg CORT did not significantly alter Ki67+ cell count. One-way ANOVA, p=0.007. *q = 3.15, p<0.05. (**F**) 40 mg/kg CORT injection led to a sustained increase in plasma CORT 30 min

*Figure 2. Continued on next page*

*Figure 2. Continued*

and 3 hr after injection. The change in plasma CORT following 5 mg/kg CORT injection was not significantly different from oil injection. Two-way ANOVA, effect of CORT dose p<0.0001. **q = 3.62 and 3.61, p<0.001, 40 mg/kg 30 min and 3 hr, respectively. (**G**) The number of BrdU-labeled newborn cells surviving 24 hr after the end of immobilization was greater in immobilized rats than controls. *p=0.03 (**H**) the number of BrdU-labeled newborn cells surviving 24 hr after CORT/oil injection was greater in rats given 40 mg/kg CORT compared to 0 mg/kg CORT. *p=0.04. (**I**) Representative images of BrdU+ cells (black arrows) in the dorsal DG (dashed outline) of control, immobilized, 0 mg/kg and 40 mg/kg CORT-injected rats. (**J**) Experimental timeline. Rats were handled for 3 days, injected with CORT or oil vehicle then perfused 3 hr later. (**K**) No difference in Ki67+ cell number was found in the adult dorsal DG with increasing CORT dose. All values are average ± SEM. Scale bar is 100 μm.

CORT did not affect the percent of proliferative cells in either high or low FGF2 media as compared to vehicle (*Figure 5B*). These data indicate that NPCs do not respond to acute CORT in isolation, but likely rely on input from another cell type in the neurogenic niche. A growing body of work indicates that astrocytes can strongly regulate NPC dynamics through secreted factors (*Song et al., 2002*), so we next tested whether astrocytes might participate in the regulation of NPCs in response to CORT. We treated primary hippocampal astrocytes cultured as described in (*McCarthy and de Vellis, 1980*) with 1 μM CORT or EtOH vehicle for 3 hr then extracted astrocyte conditioned media (ACM). When administered to isolated NPCs, CORT-treated ACM increased the percent of BrdU+ NPCs significantly over EtOH-treated coculture media (CoC) control by 1.52-fold (29.56 ± 2.81 CoC-EtOH vs 44.93 ± 2.82 ACM-1 μM CORT; *Figure 5C,D*). These findings suggest that astrocytes could mediate the in vivo increase in NPC proliferation following acute stress through secreted factors.

## Acute stress increases dorsal hippocampus FGF2 expression

Given that astrocytes secrete a variety of growth factors that support cell proliferation, we next investigated hippocampal growth factor expression in stressed rats. We quantified levels of the following growth factors previously reported to be mitogenic and/or elevated by acute stress (*Mocchetti et al., 1996*; *Molteni et al., 2001*; *Ma et al., 2009*) in the dorsal hippocampus of stressed and control rats: FGF2, brain derived neurotrophic factor (BDNF), nerve growth factor (NGF), FGF receptor 1 (FGFR1), FGFR2, FGFR3, FGFR4, vascular endothelial growth factor (VEGF), growth arrest and DNA damage–inducible 45β (GADD45β) as well as the early immediate gene, CFOS. Acute stress caused a significant increase in FGF2 mRNA and protein levels in the dorsal hippocampus (*Figure 6A–D,I*). Immobilization increased fgf2 mRNA 1.77 (±0.20) fold over control while 40 mg/kg CORT increased fgf2 mRNA by 1.70 (±0.23) fold over 0 mg/kg CORT (*Figure 6A,B*). Immobilization and 40 mg/kg CORT also significantly increased FGF2 protein levels in the dorsal hippocampus by 1.59 (±0.24) fold and 2.54 (±0.34) fold, respectively (*Figure 6C,D,I*). In contrast, bdnf exon IV mRNA levels were significantly decreased by immobilization and CORT (*Figure 6E,F*). BDNF protein level, however, was unchanged by either manipulation (*Figure 6G–I*). Acute stress did not consistently alter mRNA levels of ngf, fgfr1, fgfr2, fgfr3, fgfr4, vegf, bdnf exon IX, or gadd45β (*Figure 7A–P*). mRNA of the immediate early gene cfos was generally increased at 30 min after any manipulation, including oil injection (*Figure 7Q,R*).

## Acute stress does not increase FGF2 expression in the ventral hippocampus

We next quantified FGF2 expression in ventral DG where we previously saw no effect of stress on cell proliferation. Immobilization and 40 mg/kg CORT injections significantly increased fgf2 mRNA levels in the ventral hippocampus, similar to the dorsal hippocampus (*Figure 6J,K*). However, protein levels of FGF2 were not increased by any of the acute stressors in the ventral portion of the hippocampus (*Figure 6L–N*). These results provide intriguing correlative evidence that increased FGF2 protein levels following acute immobilization or CORT exposure could underlie the observed increase in dorsal hippocampal proliferation.

## Acute stress increases FGF2 expression in hilar astrocytes of the dorsal hippocampus

To determine whether astrocytes were the source of increased dorsal hippocampal FGF2 in vivo, we used confocal microscopy to quantify double immunohistochemical labeling for FGF2 and the astrocyte marker, glial fibrillary acidic protein (GFAP) in adult male rats treated as in *Figure 2A*. Consistent with previous reports (*Bhatnagar et al., 1997*), FGF2-immunoreactive cells were found throughout the

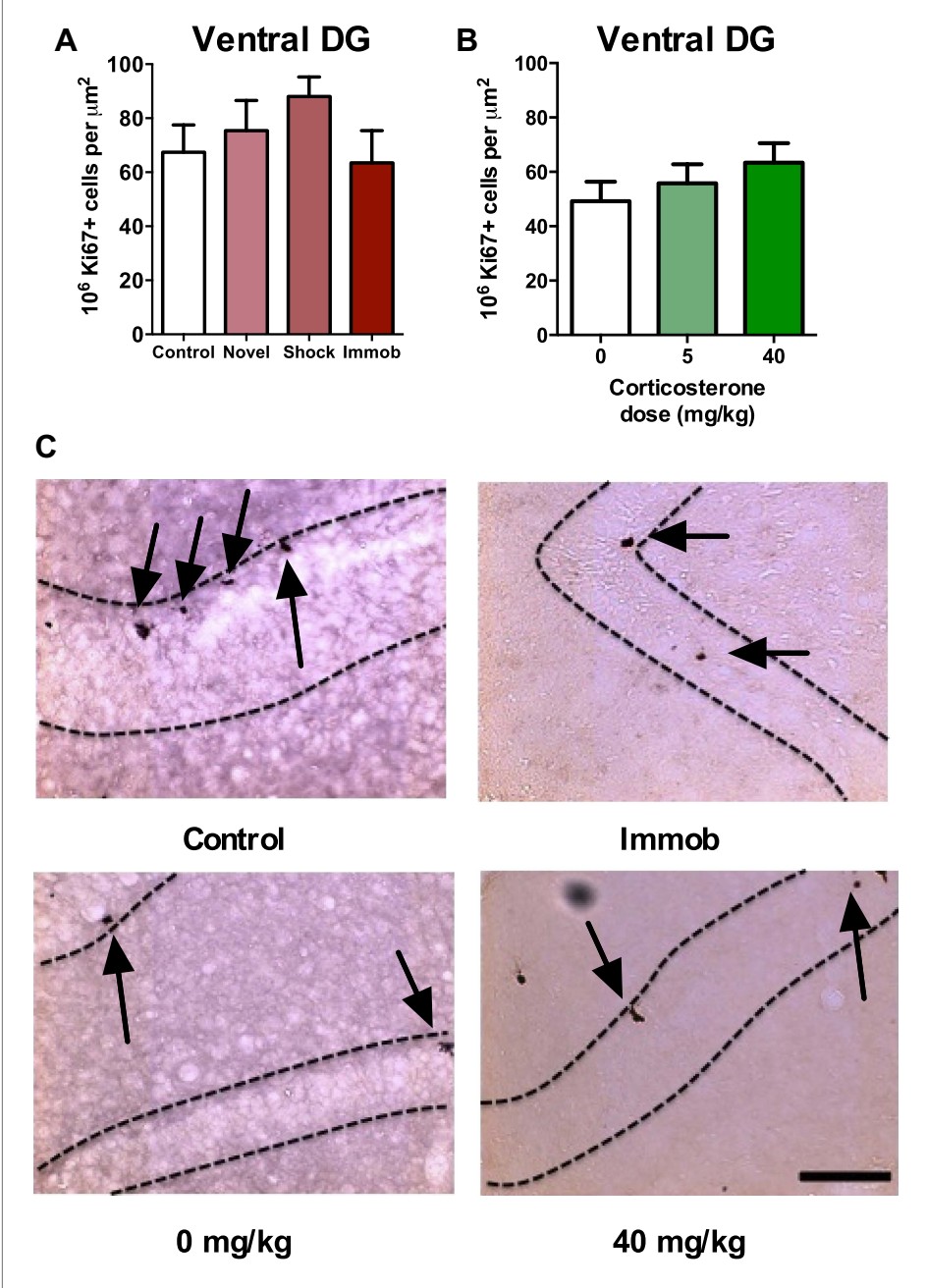

**Figure 3.** Acute stress does not increase adult cell proliferation in ventral hippocampus. (**A**) None of the stressors affected Ki67+ cell count in the ventral DG. (**B**) CORT did not affect Ki67+ cell count in the ventral DG. (**C**) Representative images of Ki67+ cells (black arrows) in the ventral DG (dashed outline) of control, immobilized, 0 mg/kg and 40 mg/KG CORT-injected rats. All values are average ± SEM. Scale bar is 100 μm.

DG and the hilus, with staining primarily in the cell body and nucleus. The DG is primarily composed of granule neurons and while almost every cell expressed FGF2, we found very few GFAP+ cells within the DG as expected. Mean optical density (with background correction) throughout the Z-stack of FGF2 expression in the DG revealed no effect of stress or CORT injection on DG FGF2 expression (*Figure 8A,D*). In the hilus, a mixture of GFAP+ and GFAP- cells was found, with almost all GFAP+ cells being FGF2+. By quantifying integrated optical density of individual FGF2+ cells that were either GFAP+ or GFAP−, we found that both immobilization and 40 mg/kg CORT injection increased FGF2

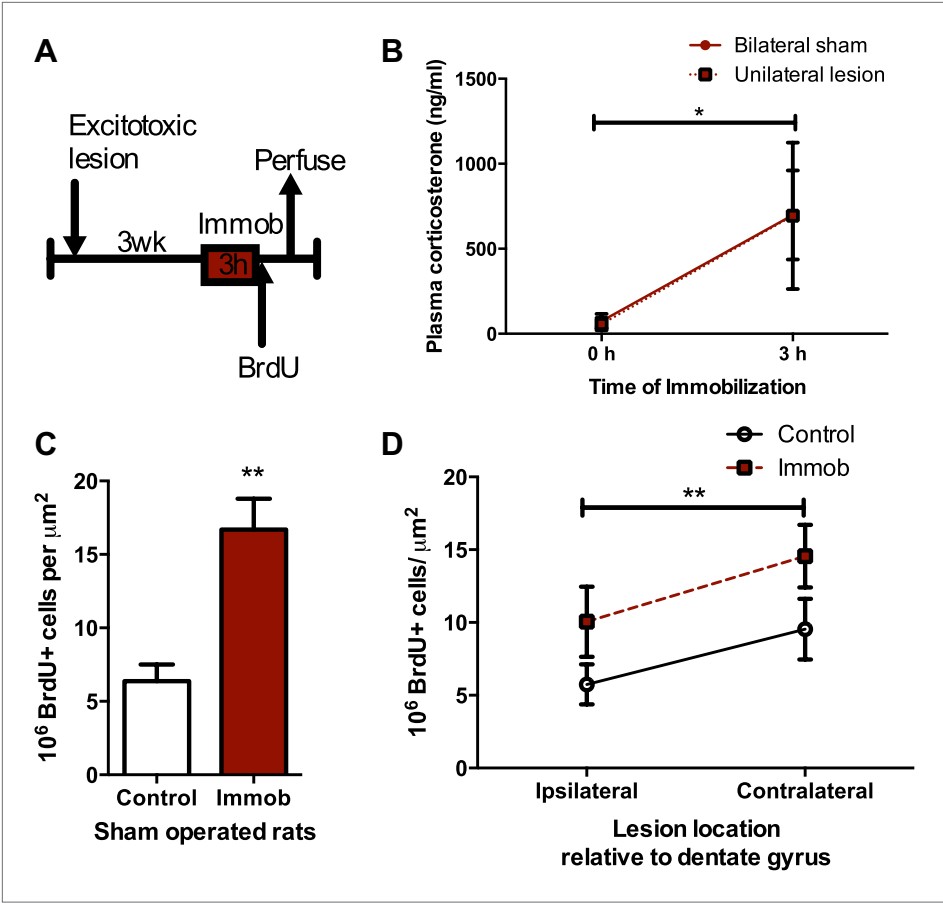

**Figure 4.** Acute stress increases cell proliferation independent of BLA input. (**A**) Experimental timeline. (**B**) Plasma CORT elevation after immobilization was similar between sham-operated and unilaterally BLA-lesioned rats. Two-way ANOVA effect of time, *p=0.04. (**C**) In sham-operated rats, acute immobilization increased the number of BrdU+ cells in the adult DG. **p=0.001. (**D**) Unilateral excitotoxic lesion of the BLA decreased the number of BrdU+ cells in the ipsilateral DG, but did not interact with stress. Two-way ANOVA, effect of lesion, **p=0.002. All values are average ± SEM.

signal in GFAP+ cells significantly over their respective control groups (*Figure 8B,D*). No effect of immobilization or CORT was observed in GFAP− cells in the hilus (*Figure 8C,D*). These findings suggest that the stress-induced increase in FGF2 levels in the DG most likely comes from neighboring astrocytes in the hilus.

## FGF2 neutralization in CORT-ACM blocks increased NPC proliferation

To determine whether FGF2 was the astrocyte-derived factor driving NPC proliferation, we next examined the role of FGF2 in the effect of CORT-treated ACM on isolated NPCs. Consistent with previous work (*Forget et al., 2006*), media from untreated astrocytes showed no detectable FGF2 (0.12 ± 0.46 pg/ml). In contrast, astrocytes treated with 1 µM CORT for 3 hr secreted 3.5 ± 0.68 pg/ml FGF2 protein (*Figure 9A*). We next tested whether levels of FGF2 as low as 4 pg/ml were sufficient to stimulate NPC proliferation. Treating with 4 pg/ml rat recombinant FGF2 caused a significant 1.39-fold increase in BrdU labeling compared to vehicle (0 pg/ml FGF2), suggesting that the levels of FGF2 present in CORT-ACM were sufficient to stimulate NPC proliferation (11.95 ± 1.22% BrdU+, 0 pg/ml; 16.65 ± 1.72% BrdU+, 4 pg/ml, p=0.04). We next investigated whether blocking FGF2 function could prevent acute stress-induced enhancement of NPC proliferation using a neutralizing antibody against FGF2 (nAb). An ELISA for rat FGF2 revealed that the nAb decreased available FGF2 over a wide range of concentrations (*Figure 9B*). The nAb did not, however, affect the availability of FGF1 (*Figure 9C*), a closely related member of the FGF family. When tested in vitro, we found that FGF2 neutralization

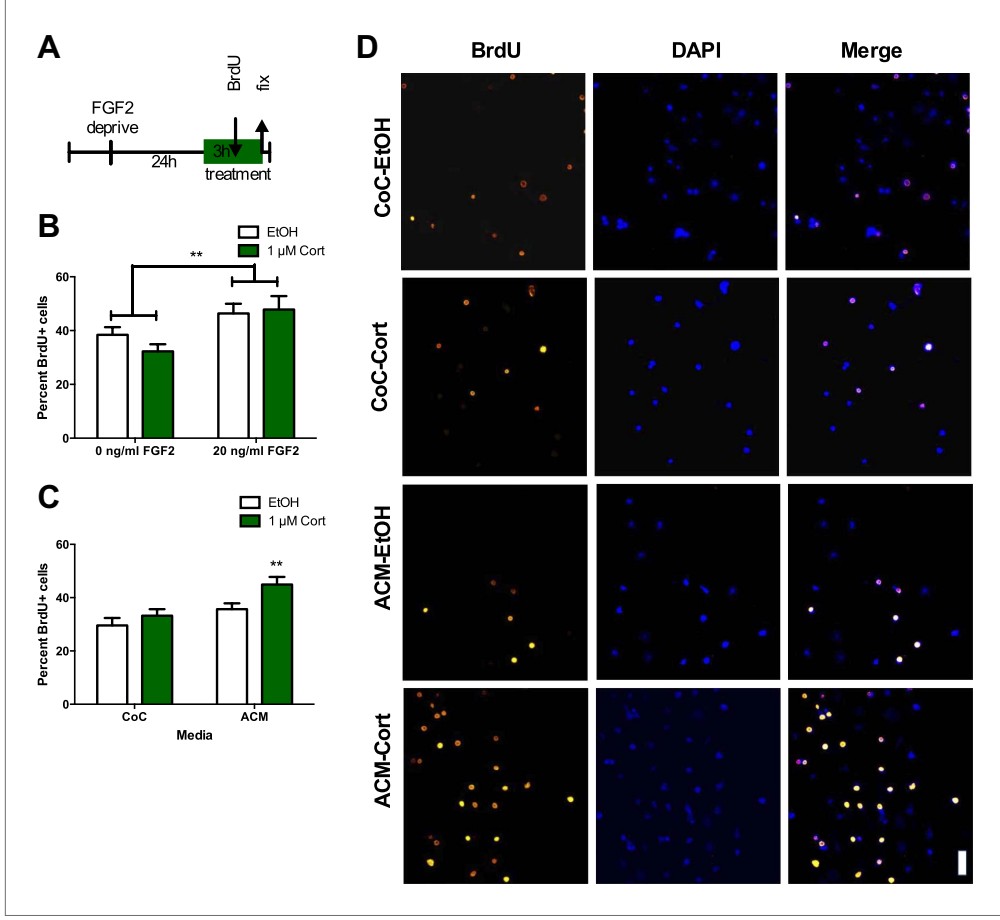

**Figure 5.** ACM from CORT-treated astrocytes increases NPC proliferation. (**A**) Experimental timeline. (**B**) Treatment of isolated hippocampal NPCs with 1 μM CORT for 3 hr did not alter the percent of proliferating BrdU+ cells compared to EtOH vehicle. 3 hr of treatment with 20 ng/ml human recombinant FGF2 increased the percent of proliferative BrdU+ cells. Two-way ANOVA, effect of FGF2, **p=0.005. (**C**) ACM was extracted from astrocytes treated with 1 μM CORT or EtOH vehicle. Treatment of NPCs with ACM from CORT-treated astrocytes increased the percent of proliferative BrdU+ cells compared to EtOH, CoC-treated control NPCs. Two-way ANOVA, effect of CORT, p=0.02; effect of media, p=0.0024. **q = 4.23, p<0.001 vs EtOH-CoC. (**D**) Representative images of NPCs treated with CoC or ACM, EtOH or CORT. BrdU+ cells are orange and DAPI is blue. All values are average ± SEM. Scale bar is 10 μm.

blocked FGF2-induced stimulation of NPC proliferation without affecting proliferation in FGF2-free conditions (*Figure 9D*). These findings suggest that the nAb blocks FGF2 signaling specifically and does not have nonspecific toxic effects on NPC growth. We then pretreated ACM from EtOH- or CORT-treated astrocytes with FGF2 nAb before exposing NPCs to the treated media. Pretreatment with the FGF2 nAb blocked the CORT-ACM induced increase in NPC proliferation (30.67 ± 3.44% BrdU positive in EtOH, no nAb vs 30.33 ± 2.59% BrdU positive in 1 μM CORT with nAb; *Figure 9E*), suggesting that increased astrocytic FGF2 is the driving signal for CORT-ACM-induced proliferation of adult hippocampal NPCs.

## Acute stress leads to a delayed enhancement of fear extinction retention

Several studies suggest that newly-born neurons play an important functional role in hippocampal memory during an immature, highly plastic phase of their development (*Kee et al., 2007*; *Deng et al., 2009*; *Kitamura et al., 2009*; *Stone et al., 2010*). However, newborn neurons in the adult rat require two or more weeks to mature and become physiologically active (*Snyder et al., 2009*), implying that

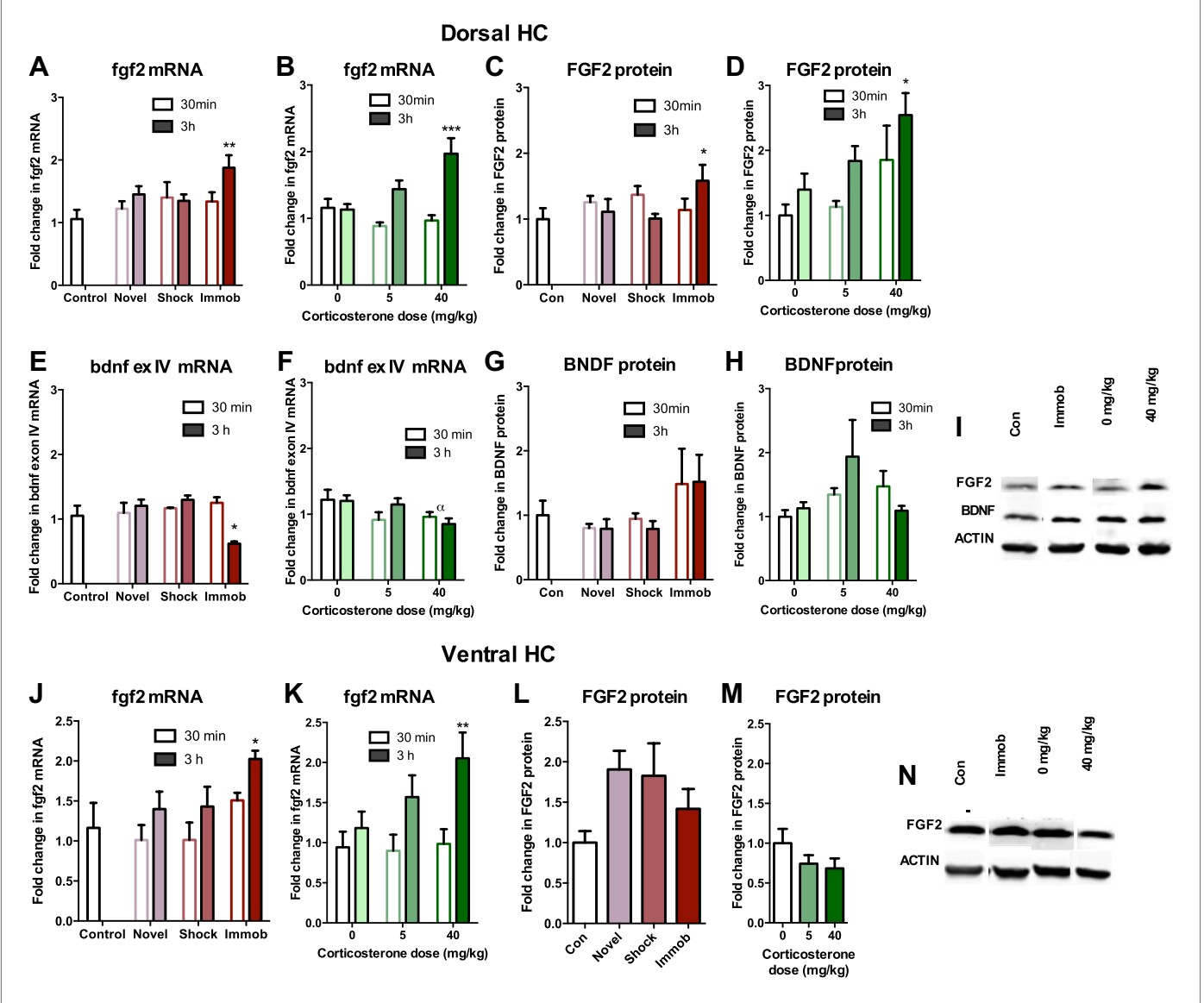

**Figure 6.** Acute stress increases FGF2 expression in dorsal hippocampus. (**A**) 3 hr of immobilization increased fgf2 mRNA expression over control in dorsal hippocampus. Other groups did not significantly differ from control. One-way ANOVA, p=0.05. **q = 3.54, p<0.01. (**B**) 40 mg/kg CORT increased fgf2 mRNA expression in the dorsal hippocampus 3 hr after CORT injection compared to 30 min after 0 mg/kg CORT injection. Other groups did not significantly differ from oil-injected controls. Two-way ANOVA, effect of CORT, p=0.03; effect of time, p<0.0001; interaction, p=0.0021. ***q = 4.34, p<0.001. (**C**) FGF2 protein levels in dorsal hippocampus increased with 3 hr of immobilization over control. Other groups did not significantly differ from control. One-way ANOVA, p>0.05. *q = 2.79, p<0.05. (**D**) FGF2 protein levels in dorsal hippocampus increased 3 hr after 40 mg/kg CORT injection compared to 30 min after 0 mg/kg vehicle injection. Two-way ANOVA, effect of CORT, p=0.01; effect of time, p=0.03. *q = 3.18, p<0.05. (**E**) 3 hr of immobilization decreased bdnf exon IV expression over control in dorsal hippocampus. Other groups did not significantly differ from controls. One-way ANOVA, p=0.0007. *q = 3.05, p<0.05. (**F**) There was an overall significant decrease in bdnf exon IV mRNA expression with increasing CORT dose in dorsal hippocampus. Two-way ANOVA, effect of CORT, *p=0.02. (**G**) BDNF protein levels in dorsal hippocampus did not change with immobilization, novel environment or shock compared to control. (**H**) BDNF protein levels did not change compared to 0 mg/kg vehicle with increasing CORT dose. (**I**) Representative western bands of FGF2, BDNF and ACTIN from the 3 hr time point in dorsal hippocampus. (**J**) 3 hr of immobilization increased fgf2 mRNA expression over control in ventral hippocampus. Other groups did not significantly differ from controls. One-way ANOVA, p=0.03. *q = 2.87, p<0.05. (**K**) 40 mg/kg CORT increased fgf2 mRNA expression in the ventral hippocampus 3 hr after CORT injection compared to 30 min after 0 mg/kg CORT injection. Other groups did not significantly differ from oil-injected controls. Two-way ANOVA, effect of time, p=0.002. **q = 3.34, p<0.01. (**L**) Immobilization, novel environment and shock did not alter FGF2 protein levels in ventral hippocampus. (**M**) CORT did not alter FGF2 protein levels in ventral hippocampus 3 hr after injection. (**N**) Representative western bands of FGF2 and ACTIN from the 3 hr time point in ventral hippocampus. All values are average ± SEM.

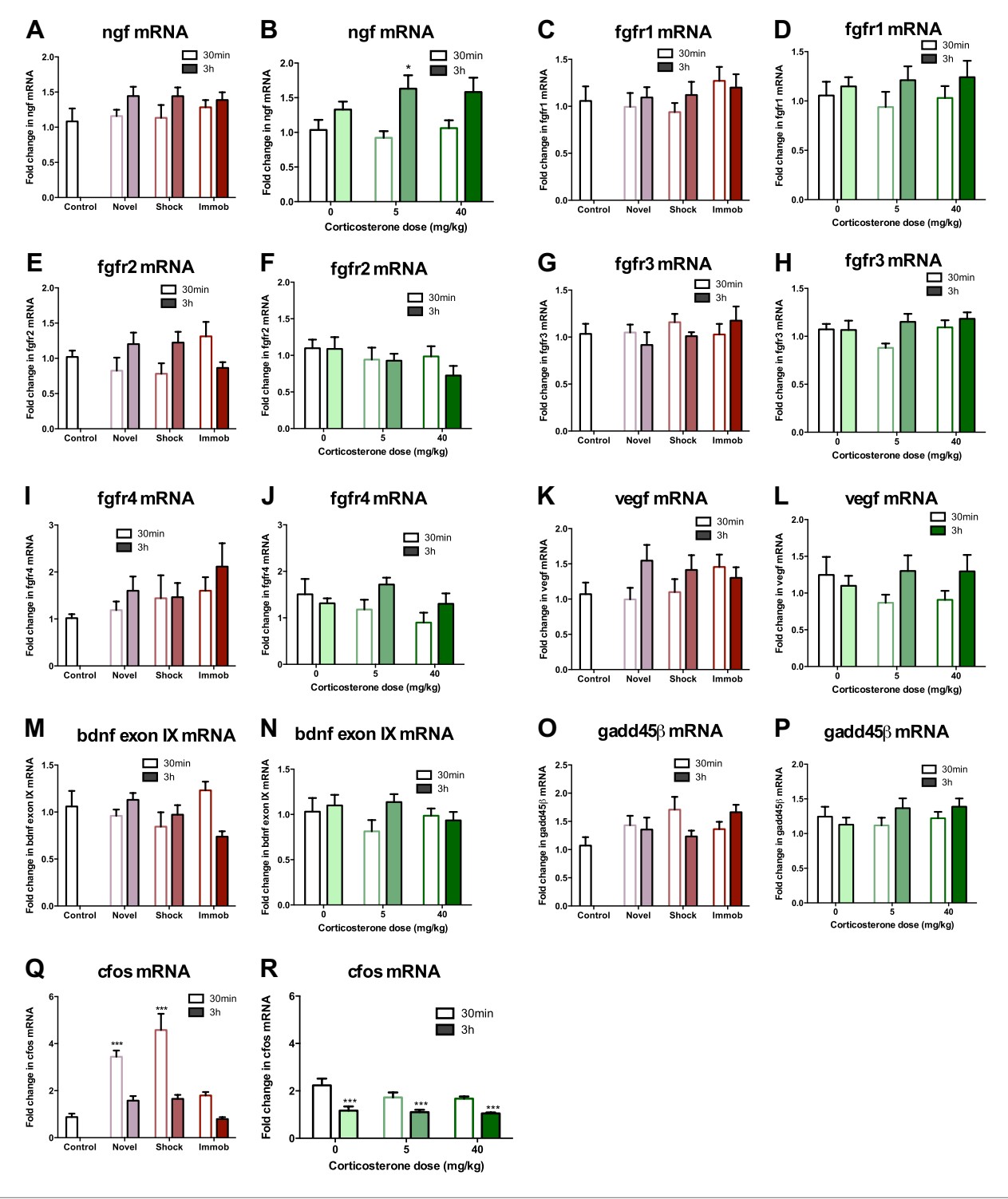

**Figure 7.** mRNA expression levels in dorsal hippocampus following acute stressors. (**A**) There was no change in ngf mRNA with novel environment, shock or immobilization. (**B**) 5 mg/kg CORT significantly increased ngf mRNA at 3 hr over 0 mg/kg CORT, 30 min *q = 2.79, p<0.05. There was no change in fgfr1 (**C**, **D**), fgfr2 (**E**, **F**), fgfr3 (**G**, **H**), fgfr4 (**I**, **J**), vegf (**K**, **L**), bdnf exon IX (**M**, **N**) Or gadd45β (**O**, **P**) mRNA expression in dorsal hippocampus. (**Q**) Exposure to novel environment or footshock increased cfos expression in the dorsal hippocampus. One-way ANOVA, p<0.0001. ***q = 5.92 and q = 8.54, novel and shock, respectively, p<0.001. All values are average ± SEM. (**R**) All injection conditions showed a decrease in cfos mRNA over time. Two-way ANOVA, effect of time, p<0.0001. ***q = 4.42, 4.68, 4.91, 0 mg/kg 3 hr, 5 mg/kg 3 hr, 40 mg/kg 3 hr, respectively, p<0.001. All values are average ± SEM.

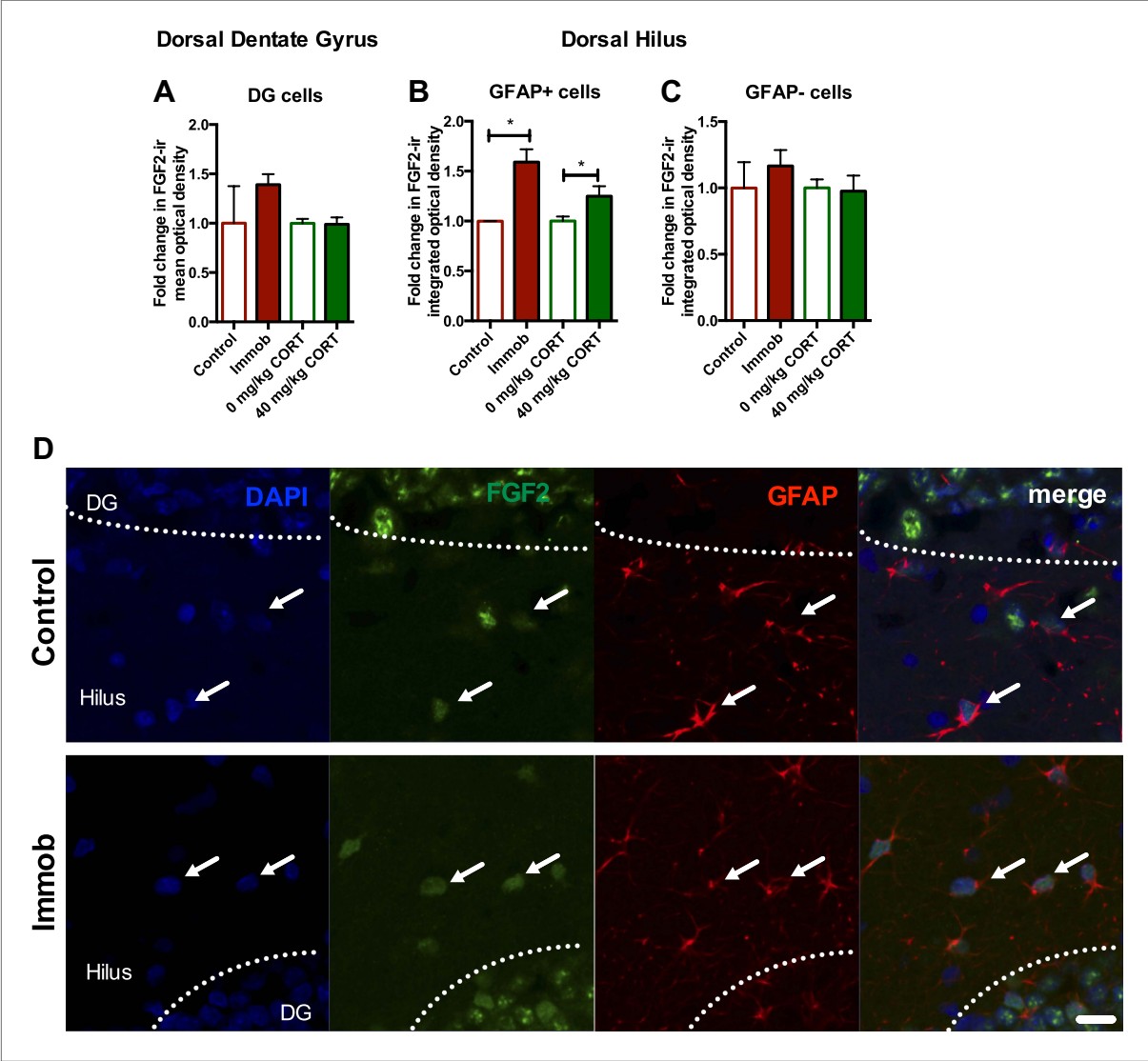

**Figure 8.** Acute stress increases FGF2 expression in GFAP+ astrocytes in the dorsal hilus. (**A**) There was no change in mean optical density of FGF2-ir in the DG following 3 hr immobilization or CORT injection. (**B**) Both immobilization and 40 mg/kg CORT injection significantly increased integrated optical density of FGF2-ir in GFAP+ cells in the hilus. One-way ANOVA, p=0.0024. *p=0.04 and 0.05, con v immob and 0 V 40 mg/kg CORT, respectively. (**C**) FGF2-ir integrated optical density in GFAP- cells of the hilus did not change. (**D**) Representative images of FGF2+ cells (green) that are GFAP+ (red; white arrows) or GFAP- in DG and hilus of a control and an immobilized rat. DAPI is blue. Scale bar = 10 μm.

the enhancement in neurogenesis we observed following acute stress should require several weeks to influence behavior. We therefore assessed hippocampal memory using a fear conditioning task either 2 days or 2 weeks after acute stress. Immobilized and control rats were given 10 unsignaled, 1 s, 1 mA shocks in a fear conditioning chamber. The next day, they received five 10 min extinction trials where they were exposed to the fear conditioning chamber without shock. On the third day of testing, they received a single 10 min extinction probe trial without shock. When rats were tested 2 days after immobilization, control and immobilized rats showed similar freezing behavior during training, extinction and the 24 hr extinction probe (*Figure 10A–C*). When rats were tested 2 weeks after immobilization, immobilized rats did not differ from controls in percent time freezing during training or during the five extinction trials (*Figure 10D,E*). During the extinction probe, however, immobilized rats froze significantly less than control rats, indicating better retention of the fear extinction (*Figure 10F*). These results suggest that an incubation time is necessary for acute immobilization to improve fear memory

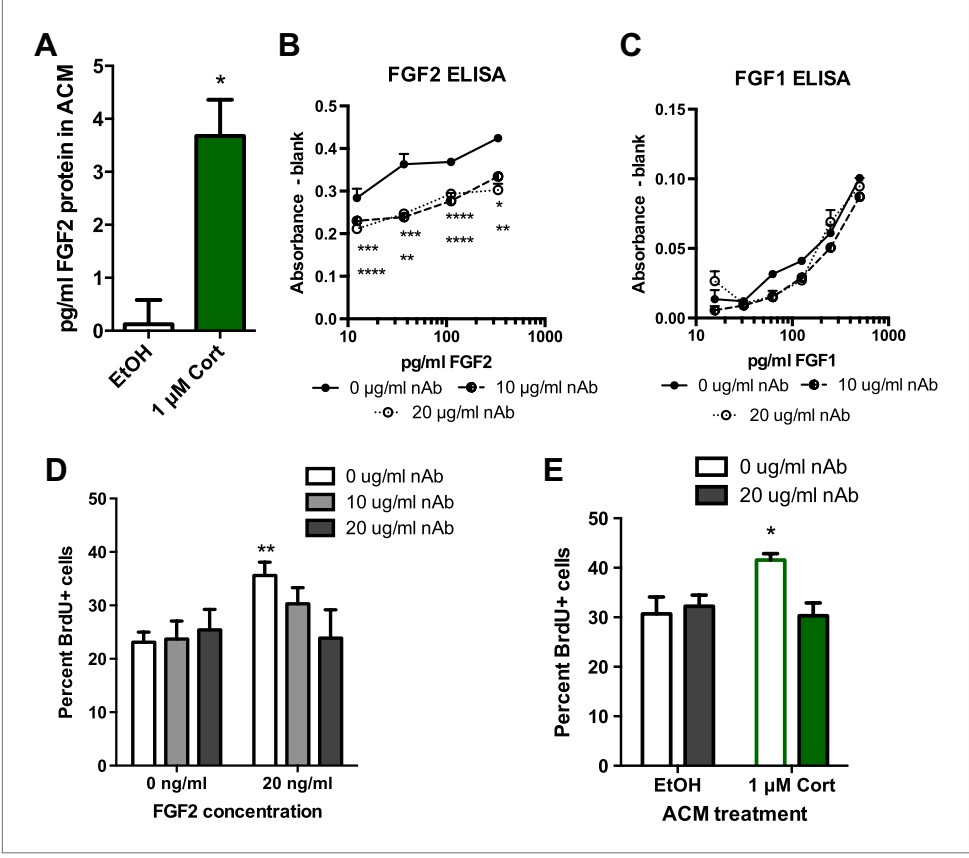

**Figure 9.** Blocking FGF2 prevents CORT-ACM induced increase in NPC proliferation. (**A**) ACM from EtOH-treated primary astrocytes had no FGF2 protein (relative to blank) while CORT-treated ACM contained 3.5 pg/ml FGF2. *p=0.01. (**B**) Availability of rat FGF2 was dramatically reduced by pretreating FGF2 protein with an FGF2 neutraliz-ing antibody. Two-way ANOVA, p<0.0001 main effects of nAb and FGF2 concentration. Post-hoc Dunnett's multiple comparison tests with 0 µg/ml as control shown for 10 µg/ml (upper row *s) and 20 µg/ml (lower row *s): *p<0.05, ** p<0.01, ***p<0.001, ****p<0.0001. (**C**) Availability of FGF1 was not affected by pretreating FGF1 protein with the same FGF2 neutralizing antibody. (**D**) Isolated NPCs were treated with the FGF2 nAb in either high (20 ng/ml) or low (0 ng/ml) FGF2 conditions. 20 µg/ml nAb effectively blocked the FGF2-induced increased in percent BrdU+ proliferating cells. Two-way ANOVA, effect of FGF2, p=0.04. **q = 3.37, p<0.01. The nAb did not affect proliferation in low FGF2 conditions. (**E**) Pretreatment with FGF2 nAb prevented the increase in percent BrdU+ proliferating NPCs caused by ACM from CORT-treated astrocytes. Two-way ANOVA, interaction, p=0.02. *q = 3.06, p<0.05. All values are average ± SEM.

and demonstrate enhanced hippocampal memory at a time when newborn neurons are highly plastic and sensitive to environmental input.

## Acute stress leads to enhanced activation of newly born neurons

To determine if the neurons born 2 weeks before fear conditioning might play a role in the enhance-ment of fear memory retention, we quantified cell fate and activation in a subset of rats perfused 1 hr after the fear extinction probe. The number of surviving BrdU+ cells (*Figure 10G*) and the percent of BrdU+ cells expressing the immature neuronal marker doublecortin (DCX+) or the astrocytic marker GFAP+ (*Figure 10H*) were similar in immobilized and control rats. However, immobilized rats had sig-nificantly more immature (DCX+) cells expressing the immediate early gene cfos than control rats (1.57 ± 0.52% cfos+ vs 5.39 ± 1.40% cfos+; *Figure 10I,J*), suggesting greater activation of immature neurons in immobilized rats. These results indicate enhanced utilization of the highly plastic pool of new neurons born around the time of an acute stressor and suggest that stress-stimulated proliferation may support later memory benefits.

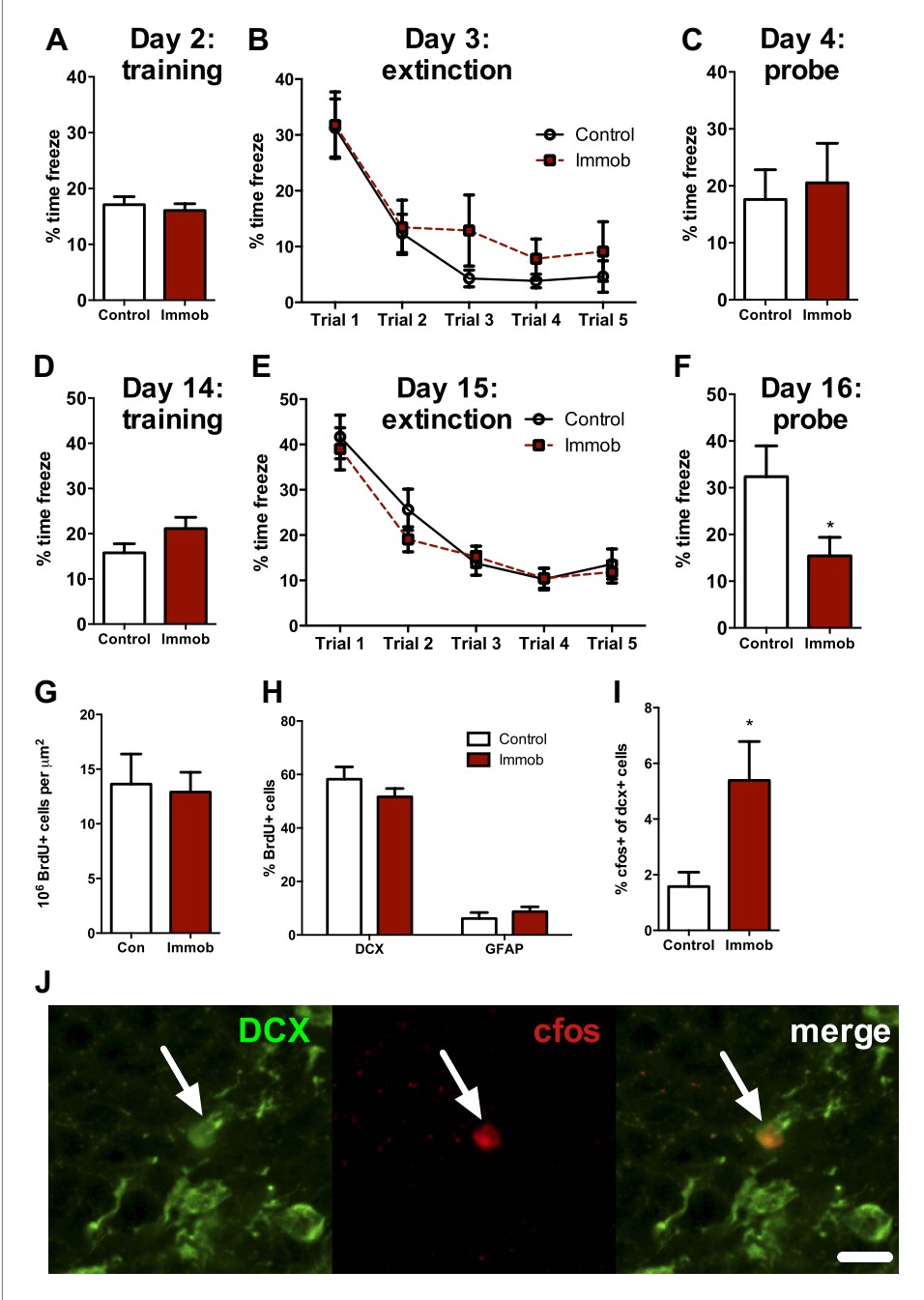

**Figure 10.** Acute stress causes delayed enhancement of contextual fear extinction retention. Acute immobilization 2 days prior to contextual fear conditioning did not change percent time freezing during training (**A**), extinction (**B**) or 24-hr extinction probe (**C**) compared to control. Two-way ANOVA for extinction, effect of trial, $p < 0.0001$. Acute immobilization 2 weeks prior to contextual fear conditioning did not change percent time freezing during training (**D**) or extinction (**E**) compared to control. Two-way ANOVA for extinction, effect of trial, $p < 0.0001$. (**F**) Immobilized rats froze significantly less than controls in the 24-hr extinction probe. *$p = 0.04$. (**G**) The number of surviving BrdU+ cells in the dorsal DG of immobilized rats did not differ from controls. (**H**) Immobilization does not alter the percent of BrdU+ cells co-expressing doublecortin (DCX) or glial fibrillary acidic protein (GFAP). (**I**) The percent of DCX+ immature neurons expressing cfos was greater in immobilized rats compared to controls. *$p = 0.02$. All values are average ± SEM.

## Discussion

The present study demonstrates that acute stress or exposure to the stress hormone CORT induces an increase in proliferation of hippocampal NPCs via increased secretion of astrocytic FGF2. This increase in proliferation is correlated with selective activation of the hyper-plastic newborn neurons and enhanced retention of fear extinction 2 weeks after the stressor. Taken together, these findings suggest a beneficial role for acute stress on hippocampal plasticity. Consistent with our findings, previous studies using similar stress paradigms show stress-induced enhancements in memory consolidation and growth factor expression (*Roozendaal et al., 1999*; *Molteni et al., 2001*; *Roozendaal et al., 2009b*). Acute stress may also support hippocampal LTP, another factor implicated in stimulating adult hippocampal neurogenesis (*Korz and Frey, 2005*). Notably, other manipulations that increase stress hormone secretion, such as exercise, sexual experience and mild immune challenge, similarly stimulate adult neurogenesis (*Kronenberg et al., 2006*; *Wolf et al., 2009*; *Leuner et al., 2010*; *Buwalda et al., 2012*). Recent work in adult squirrel monkeys also shows that coping with intermittent social stress through multiple pair separations and new pair formations both stimulates adult hippocampal neurogenesis and improves hippocampal memory (*Lyons et al., 2010*). These studies, along with our findings, evoke the interesting possibility that acute stress may be beneficial for brain health in general and hippocampal plasticity in particular.

Previous studies of adult NPC response to acute stress have yielded mixed results. While some initial work suggested that acute stress causes a decrease in NPC proliferation (*Cameron and Gould, 1994*; *Gould et al., 1997*), more recent publications have not replicated these findings (*Thomas et al., 2006*; *Thomas et al., 2007*; *Dagyte et al., 2009*; *Hanson et al., 2011*). Notably, while we found that acute CORT enhanced proliferation in well-handled, habituated rats, we did not find any effect of acute CORT on neurogenesis in rats that were not habituated to injection. Our findings therefore demonstrate that factors present before the stressor begins may alter the effect of acute stress, perhaps by changing the susceptibility of controls or stressed animals to manipulation. Not surprisingly, a rich literature exists regarding the role of handling on altering the rodent stress response (*Korosi and Baram, 2010*). Such experimental differences could explain discrepancies in the literature.

Previous research concerning the long-term effects of acute stress in rodents focuses primarily on models of traumatic stress and the resultant PTSD-like symptoms. Most prominently, the single prolonged stress model of PTSD uses three acute stressors in series (restraint, forced swim and ether) and results in delayed deficits in fear extinction, as well as enhanced anxiety (*Knox et al., 2012a*, *2012b*). However, if any one of the three components of this stressor protocol are eliminated, deficits are no longer evident (*Knox et al., 2012a*), suggesting that stressor effects depend on the severity and length of the stressor, perhaps following the inverted U function previously described (*Lupien and McEwen, 1997*).

Previous work in primates has provided additional evidence for an inverted U relationship between stress and neural function. In multiple non-human primate species, for example, mild early life stress can lead to resilience against stress later in life, a phenomenon referred to as stress inoculation (*Lyons and Parker, 2007*; *Katz et al., 2009*; *Parker and Maestripiero, 2011*). In contrast, more severe early life stress can have the opposite outcome, increasing stress vulnerability later in adulthood. When combined with our current findings, these studies fit well with the proposed inverted U function for stress effects on brain health (*Lupien and McEwen, 1997*; *Salehi et al., 2010*; *Luksys and Sandi, 2011*). At high, traumatic levels, acute stress may result in maladaptive pathology (e.g., PTSD-like symptoms in the single prolonged stress model). At more moderate levels, though, such as immobilization in well-handled rats, acute stress may actually enhance function. Future research will be needed to more precisely define the limits of stimulating vs detrimental acute stress.

Many acute-stress-induced changes in hippocampal plasticity rely on functional input from the BLA. For example, while stress enhances hippocampal memory consolidation and LTP, lesion of the BLA blocks these enhancements (*Quirarte et al., 1997*; *Roozendaal et al., 2009a*, *2009b*). We have recently reported that hippocampal cell proliferation and the activation of the newborn neurons in a fear-conditioning paradigm both depend on BLA neural input (*Kirby et al., 2012b*). In the current study, we found that while BLA lesion suppressed hippocampal cell proliferation as we have shown before, it did not affect the acute stress-induced increase in neurogenesis. These findings imply that acute stress regulation of neurogenesis may not rely on the same systems-level in vivo circuitry that mediates acute stress regulation of memory consolidation and LTP. In support of this hypothesis, we were able to model CORT-induced NPC proliferation in isolated NPCs. We found that while NPCs did not respond to acute CORT treatment independently in vitro, they did proliferate more in response to

conditioned media from CORT-treated primary astrocytes, suggesting a role for secreted factors from local astrocytes in mediating stress effects on NPCs.

The dynamic role of astrocytes in facilitating neuronal function through secreted factors has gained much recognition (*Eroglu and Barres, 2010*). In addition to aiding in synaptic glutamate recycling, astrocytes secrete several factors such as thrombospondins (*Eroglu and Barres, 2010*), Hevin, SPARC (*Kucukdereli et al., 2011*) and glypicans (*Allen et al., 2012*) that regulate synaptic formation and function in mature neurons (*Eroglu and Barres, 2010*). Astrocytes express the glucocorticoid receptor, GR, and when they are exposed to high levels of GCs, GC-bound GR translocates to the nucleus and enhances FGF2 gene transcription (*Molteni et al., 2001*; *Gubba et al., 2004*; *Unemura et al., 2012*). FGF2 is a potent and necessary proliferative factor in adult NPCs (*Chipperfield et al., 2002*). We found that acute stress stimulated FGF2 expression in the dorsal hippocampus and in primary hippocampal astrocytes. We further showed that in the DG and hilus, the enhancement in FGF2 levels following stress is largely restricted to GFAP+ astrocytes. Notably, changes in FGF2 protein levels following acute stress closely paralleled the neurogenic response; that is, FGF2 levels were increased only in stress conditions that also stimulated neurogenesis. Neutralizing the astrocyte-secreted FGF2 prevented enhanced proliferation in cultured NPCs. These findings suggest a novel role for astrocytes in supporting hippocampal plasticity in response to an environmental stressor through secreted FGF2. Further research will be required to fully dissect the molecular mechanisms by which stress induces FGF2 secretion from astrocytes.

Newly-born neurons are implicated in numerous hippocampal memory functions (*Aimone et al., 2010*), particularly contextual fear conditioning. Contextual fear conditioning activates immature neurons (*Kirby et al., 2012b*) and when new neurons are selectively knocked down, fear extinction is impaired (*Deng et al., 2009*; *Stone et al., 2010*). Sahay et al. have further shown that prolonging survival of newly-born neurons through targeted knockdown of apoptotic pathways enhances discrimination between similar contexts in a fear conditioning task (*Sahay et al., 2011*). We found that 2 weeks, but not 2 days, after acute stress, immobilized rats showed enhanced contextual fear extinction retention compared to controls. We also found that immobilized rats had greater activation of newly born neurons in response to fear extinction recall. This time window coincides with the period in which the newborn cells are hyperplastic and more likely to be recruited to active circuitry (*Kee et al., 2007*; *Deng et al., 2009*; *Kirby et al., 2012b*). Given that immobilized rats had a similar number of newborn neurons as controls, these data suggest that immobilized rats better utilize the pool of immature neurons, possibly contributing to their enhanced memory. However, this connection does not preclude the potential contribution of other aspects of hippocampal plasticity to the enhanced memory. Future research will be necessary to fully determine the role of immature neurons vs existing circuitry in enhancement of hippocampal memory.

A recent hypothesis posits that the dorsal and ventral hippocampus support different behavioral functions, with the dorsal region being important for spatial and declarative memory while the ventral region facilitates affective regulation (*Fanselow and Dong, 2010*). This division may also apply to neural stem cell regulation within the DG of each region where chronic stress preferentially effects the ventral over the dorsal hippocampus, perhaps reflecting the negative affective consequences of chronic stress (*Tanti et al., 2012*). In the present study, we found that acute stress or stress hormone exposure increased neurogenesis and FGF2 expression in the dorsal, but not in the ventral, hippocampus. Moreover, the memory benefits observed 2 weeks after stress were in contextual fear extinction, a task that relies on the kind of spatial memory hypothesized to depend on the dorsal DG (*Fanselow and Dong, 2010*). This selective involvement of the dorsal hippocampus implies that our acute stressor model functions as a cognitive stimulant for the declarative domains of hippocampal function rather than as a modulator of emotional responsivity.

The brain's response to acute stress can define the line between life-saving adaptation and long-term pathology. The current study suggests that moderate, acute stress may stimulate heightened brain plasticity via increased neurogenesis. These findings have important implications for understanding adaptive vs pathological responses to stress.

## Materials and methods

### Animals

Adult male Sprague-Dawley rats (Charles River) were pair-housed on a 12 hr light dark cycle with lights on at 07:00 hr. Rats were allowed to acclimate to the animal facility for 1 week before handling began. All animal procedures were approved by the UC Berkeley Animal Care and Use Committees.

## Stressors

Novel environment, footshock, immobilization and control rats were all handled for 5 days. On the sixth day, they were exposed to a stressor or left undisturbed in the case of controls. The novel environment and footshock exposure both lasted 30 min and occurred in fear conditioning chambers described in the 'Fear conditioning'. For footshock, rats were exposed to 1 mA, 1 s duration unsignaled shock 30 times. Rats were returned to their home cage after the 30 min exposure until the time of sacrifice. Immobilized rats were confined for 3 hr in decapicone bags (Braintree Scientific, Braintree, MA).

## Corticosterone injection

For most experiments, rats were handled for 2 days, given a subcutaneous (SC) needle stick on day 3 and then injected with sesame oil (SC) for 2 days (days 4 and 5). On the sixth day, rats received 0, 5 or 40 mg/kg corticosterone (SC, Sigma, St. Louis, MO) suspended in sesame oil. CORT was either at 5 mg/ml or 40 mg/ml such that rats receiving 5 or 40 mg/kg corticosterone received equal volumes of oil relative to body weight. In the case of the rats not pre-injected, rats were handled for 3 days and then injected with 0, 5 or 40 mg/kg corticosterone as above on the fourth day.

## BrdU injection

5-Bromo-2′-deoxyuridine (BrdU, Sigma) was dissolved in physiological saline. Rats were injected with BrdU (intraperitoneally, 200 mg/kg) 3 hr after the beginning of the stressor or injection unless otherwise noted.

## BLA lesion

Excitotoxic lesions of the BLA were performed using unilateral stereotaxic infusion of N-methyl-d-aspartate (Sigma) as per (*Kirby et al., 2012a*, *2012b*). Coordinates for BLA infusion were: −2.8 mm anterior/posterior (A/P), ±5.1 mm medial/lateral (M/L) relative to bregma; −6.8 mm (2 min) and −6.5 mm (1 min) relative to dura. Following 3 weeks of recovery, during which rats were handled regularly, rats were immobilized for 3 hr (n = 7 each, sham and unilateral lesion) or left undisturbed in their home cage (n = 7 each, sham and unilateral lesion, respectively). Tail vein blood samples were taken at the beginning and end of immobilization for plasma corticosterone quantification. At the end of immobilization, all rats received one injection of 100 mg/kg BrdU and were perfused 2 hr later.

## Immunohistochemical staining

Rats were anesthetized with Euthasol euthanasia solution and transcardially perfused with ice cold 0.1 M phosphate buffered saline (PBS) followed by 4% paraformaldehyde in 0.1 M PBS. Brains were post-fixed for 24 hr at 4°C, equilibrated in 30% sucrose in 0.1 M PBS and then stored at −20°C. Immunostaining was performed on a 1 in 6 series of free-floating 30 µm cryostat sections.

Ki67 staining was done for sections from control (n = 10), novel environment (n = 6), footshock (n = 4), immobilized (n = 6), 0 mg/kg (n = 6), 5 mg/kg (n = 6) and 40 mg/kg (n = 6) CORT-injected rats as per (*Kirby et al., 2012b*) with a few additions. Sections were antigen-retrieved using 10 mM citrate buffer, pH 8.0 at 95°C for 20 min prior to peroxidase blocking and the primary antibody used was rabbit anti-Ki67 (1:500; Novus, St. Louis, MO).

BrdU staining was done as per (*Kirby et al., 2012b*) for sections from control (n = 6), immobilized (n = 5), 0 mg/kg (n = 6), and 40 mg/kg (n= 5) rats who were perfused 24 hr after the BrdU injection, which occurred 3 hr after the start of the stressor. Sections from control (n = 10) and immobilized (n = 8) rats who were perfused 2 weeks after immobilization/BrdU injection, 1 hr after the final fear extinction probe (see 'Fear conditioning'), were also stained for BrdU using the same procedure. All stained BrdU and Ki67 sections were mounted on gelatin-coated slides, dehydrated in alcohols and coversliped with permount.

Double immunohistochemical labeling for FGF2 quantification in GFAP+ cells was done on sections from control (n = 2), immobilized (n = 6), 0 mg/kg CORT injected (n = 6) and 40 mg/kg CORT injected (n = 6) rats as per (*Kirby et al., 2012b*) with the following deviations. Primary antibodies were rabbit anti-FGF2 (1:500; Abcam, Cambridge, UK), and mouse anti-GFAP (1:1000; Cell Signalling, Danvers, MA). Secondary antibodies were AlexaFluor 488 donkey anti-rabbit (1:200; Invitrogen, Carlsbad, CA) and Cy3 donkey anti-mouse (1:200; Jackson ImmunoResearch, West Grove, PA). Following secondary incubation and rinsing, sections were mounted on gelatin-coated slides and coverslipped with Vectashield mounting medium with DAPI (Vector, Burlingame, CA).

Triple immunohistochemical labeling for cell fate analysis was done on sections from control (n = 6) and immobilized (n = 6) rats 2 weeks after immobilization/BrdU injection, after the final fear extinction

probe (see 'Fear conditioning') as per (*Kirby et al., 2012b*) with a few exceptions. Primary antibodies were goat anti-DCX (1:200; Santa Cruz Biotechnology, Dallas, TX), mouse anti-GFAP (1:100; Cell Signalling) and rat anti-BrdU (1:500; Abcam). Secondary antibodies were AlexaFluor 594 anti-goat, AlexaFluor 647 anti-mouse and biotin anti-rat (1:500; Jackson ImmunoResearch). Tertiary antibody was Streptavidin Alexa Fluor 488 (1:1000; Jackson ImmunoResearch). Double immunohistochemical staining for DCX and cfos was performed similarly, (n = 5 control, n = 7 immobilized) with goat anti-DCX as above and mouse anti-cfos (1:50; Santa Cruz Biotechnology). Secondary antibodies were as above for anti-goat and AlexaFluor 647 anti-mouse (1:500; Jackson ImmunoResearch). Sections were mounted on gelatin-coated slides and coversliped with DABCO antifading medium.

## Quantification of Ki67+ and BrdU+ cells

Ki67- and BrdU-positive cells were counted in the dorsal and ventral dentate gyrus and subgranular zone using a 40× air objective (Zeiss, Oberkochen, Germany). The area sampled was calculated using StereoInvestigator software (MicroBrightfield, Williston, VT) and used to calculate the number of positive cells per micrometer (*Lupien and McEwen, 1997*).

## Confocal analysis

For quantification of FGF2 immunoreactivity, 18 μm Z-stacks of 1 μm slices in the dorsal DG and hilus were acquired using a 20× air objective. Mean DG optical density was measured in ImageJ software using the summed Z-stack of FGF2 immunoreactivity. Mean optical density of FGF2- areas of tissue were subtracted from the DG intensity value to correct for background. Integrated optical density of GFAP+ and GFAP- negative cells in the hilus were determined by confirming GFAP expression in the Z-stack and then acquiring integrated optical density from the summed Z-stack of each individual cell. 51 to 97 GFAP+ cells and 16 to 57 GFAP- cells were sampled per rat. To quantify cFos expression in new neurons, BrdU positive cells were located in the dorsal dentate gyrus for each animal using a 40× oil objective and assessed in z-series of <1.0 μm slices to determine if other markers (DCX, GFAP) were co-expressed. Confocal images were captured on a Zeiss 510 META/NLO confocal microscope with a 40× oil objective.

## Real time quantitative PCR

Rats were lightly anesthetized with isoflurane and rapidly decapitated 30 min or 3 hr after the beginning of their respective stressors (n = 6/grp). Bilateral hippocampi were dissected and flash frozen in liquid nitrogen. Trunk blood was collected for plasma corticosterone quantification. One hippocampus per rat was used for mRNA expression quantification. The other was used for western blot analysis, right and left side being counterbalanced among groups. Real-time reverse transcriptase PCR was run on Trizol-extracted RNA with primers detailed in *Table 1*.

Primer sequences were designed using Primer1 software and checked for specificity using BLAST. Extracted RNA was treated with DNase (DNA-free, Ambion, Carlsbad, CA), and two-step PCR was used, following manufacturer instructions for iScript cDNA synthesis kit (BioRad, Hercules, CA) and then iQ SYBR Green Supermix. Samples were run in a BioRad IQ5 real-time PCR machine. After the PCR was complete, specificity of each primer pair was confirmed using melt curve analysis. Ct values were determined using PCR miner (*Zhao and Fernald, 2005*) and normalized to the reference ribosomal RNA, RPLP. Fold change in mRNA expression is relative to no stress control rats.

## Western blot

Protein was extracted by homogenizing in RIPA buffer with protease inhibitor (1:100; Calbiochem, Billerica, MA) and phosphatase inhibitor (1:10; Roche, Basel, Switzerland). Following 30 min incubation on ice, samples were centrifuged at 12,000×*g* for 30 min at 4°C. The extracted protein was stored at −80°C. Total protein content was assessed using a BCA kit (Pierce, Waltham, MA). Samples were diluted 1:1 in laemmli buffer (Biorad) + 5% β-mercaptoethanol (Fisher, Waltham, MA) and run on 4–20% Mini-PROTEAN TGX gels (Biorad) at 100 V for 1.5 hr in 1× Tris-glycine-SDS buffer. They were then transferred to nitrocellulose membrane (Biorad) at 100 V for 1 hr in 1× Tris-glycine-SDS buffer with 20% methanol. Membranes were blocked for 1 hr with 5% milk in 0.1 M Tris buffered saline with 1% Tween-20 (Fisher)(TBS-t). Membranes were incubated in primary (rabbit anti-FGF2, 1:100; Abcam; mouse anti-actin, 1:10,000; Roche; rabbit anti-bdnf, 1:500; Abcam) in blocking solution overnight at 4°C. The next day, membranes were rinsed three times with TBS-t then incubated in secondary (LiCor [Lincoln, NE] IRDye 680LT Donkey anti-mouse or LiCor IRDye 800CW Donkey anti-rabbit, 1:20,000) for 1 hr. After three final rinses, membranes were visualized using a LiCor Odyssey scanner. The correct

**Table 1.** List of primers

| Gene | Direction | Sequence |
|---|---|---|
| bdnfexonIV | + | GGAGTGGAAAGGGTGAAACA |
| | − | GGATTCAGTGGGACTCCAGA |
| bdnfexonIX | + | GAGAAGAGTGATGACCATCCT |
| | − | TCACGTGCTCAAAAGTGTCAG |
| cfos | + | GGCAAAGTAGAGCAGCTATCTCCT |
| | − | TCAGCTCCCTCCTCCGATTC |
| fgf2 | + | CGGTACCTGGCTATGAAGGA |
| | − | CTCCAGGCGTTCAAAGAAGA |
| fgfr1 | + | ACCTGAGGCATTGTTTGACC |
| | − | GTGAGCCACCCAGAGTGAAT |
| fgfr2 | + | GGCCTCTCTGAATGCTAACG |
| | − | ACGAGACAATCCTCCTGTGG |
| fgfr3 | + | TCTGGTCCTTTGGTGTCCTC |
| | − | TGAGGATGCGGTCTAAATCC |
| fgfr4 | + | GTGGCTGTGAAGATGCTGAA |
| | − | GAGGAATTCCCGAAGGTTTC |
| gadd45β | + | GTCACCTCCGTCTTCTTGGA |
| | − | GAGGCGGTGGGACTTACTTT |
| ngf | + | GGACGCAGCTTTCTATCCTGG |
| | − | CCCTCTGGGACATTGCTATCTG |
| rplp | + | CCAAAGGTTTGGGAGAACAA |
| | − | GGGTCATGGCATAGAGCAAT |
| vegf | + | GAGGAAAGGGAAAGGGTCAAA |
| | − | CACAGTGAACGCTCCAGGATT |

band size was found relative to a LiCor IRDye (680/800) protein marker ladder. All bands were quantified using LiCor Odyssey software, corrected for background and expressed relative to their corresponding actin band. Fold change in protein expression was then calculated relative to no stress control.

## Plasma corticosterone sampling

All blood samples were centrifuged at 2000×g for 15 min and plasma was extracted and stored at −20°C until assayed. Corticosterone was measured using a Corticosterone EIA kit (Enzo Life Sciences, Farmingdale, NY).

## Culturing of hippocampal NPCs

Isolation of neural stem/progenitor cells from adult rodents are described in detail in (*Gage, 2000*). Progenitors used in these experiments were purchased from Millipore (Billerica, MA; SCR022). Cells were cultured under standard conditions (37°C, 5% $CO_2$) on poly-ornithine (Sigma) and laminin (Invitrogen) coated plates in N2-supplemented (Invitrogen) Dulbecco's modified Eagle medium (DMEM)/F-12 (1:1) (Invitrogen) with 20 ng/ml recombinant human FGF-2 (PeproTech, Rocky Hill, NJ).

## Culturing of primary hippocampal astrocytes

Primary astrocyte cultures were prepared from P1–2 day old Sprague Dawley rat pup hippocampi using the method described by McCarthy and Vellis (*McCarthy and de Vellis, 1980*). Briefly, hippocampi were dissected in ice-cold media, chopped and digested using papain from papaya latex extract (Sigma) in HBSS (Invitrogen) for 20 min at 37°C. Papain was inactivated using 10% horse serum, cells were centrifuged for 1 min at 350×g and resuspended in HBSS and triturated by passing through serological and flame-polished pipettes of progressively smaller bores. Cells were then plated in

DMEM (Invitrogen) supplemented with 10% fetal bovine serum (Axenia BioLogix, Dixon, CA) and 1% Penicillin/Streptomycin (Invitrogen) at a density of $3 \times 10^6$ in T75 flasks. After reaching confluency, flasks were shaken on an orbital shaker at 225 rpm for 2 hr at 37°C. Cells were then washed 5× with warm PBS to remove suspended microglia. Astrocytes were then trypsinized and re-plated in 100 mm dishes. 24 hr after plating, astrocytes were treated with 1 µM CORT or equivalent volume of EtOH vehicle for 3 hr. ACM was then collected, filtered with a 0.2 µm sterile filter and stored at −20°C.

### Cell treatment for BrdU-labeling

In all studies, NPCs were FGF2 deprived for 24 hr then treated for 3 hr with the appropriate media. They were then pulsed with 30 µM BrdU and fixed 2 hr later with 4% paraformaldehyde for 10 min. Cell treatments were: 0 ng/ml FGF2 (+EtOH, n = 6; +CORT, n = 5), 20 ng/ml FGF2 (+EtOH, n = 5; +CORT, n = 5), CoC media (+EtOH, n = 6; +CORT, n = 6), ACM (+EtOH, n = 6; +CORT, n = 6). Treatments for the rat recombinant FGF2 experiment were 0 pg/ml FGF2 (n = 8) and 4 pg/ml FGF2 (n = 7).

### Immunocytochemical BrdU staining and quantification

Fixed cells were rinsed with 0.1 M PBS, denatured in 1 N HCl at 37°C, rinsed and blocked in 5% normal donkey serum, 0.3% triton-100 in PBS. Cells were then incubated overnight at 4°C in mouse anti-brdu (1:500; BD Biosciences, Franklin Lakes, NJ) in 2% normal donkey serum in PBS. Cells were then rinsed and incubated in Cy3 anti-mouse (1:500; Jackson Immunoresearch) in 2% normal donkey serum in PBS, rinsed, counterstained with DAPI (1:20,000 in PBS) and coverslipped with DABCO anti-fading medium. BrdU+ and DAPI+ cells were counted in randomly sampled sites within each well using StereoInvestigator software (Microbrightfield) and a 20× air objective (Zeiss).

### FGF2 neutralization

CoC media or ACM from primary astrocytes treated with EtOH or CORT was incubated with neutralizing FGF2 antibody (Millipore) for 1 hr at 37°C prior to use on NPCs. For nAb dose testing, NPCs were treated with 0 ng/ml FGF2 (0 µg/ml nAb, n = 8; 10 µg/ml nAb, n = 5, 20 µg/ml nAb, n = 3) or 20 ng/ml FGF2 (0 µg/ml nAb, n = 8; 10 µg/ml nAb, n = 7, 20 µg/ml nAb, n = 8). For ACM treatment with the FGF2 nAb, all n = 6.

### FGF2 ELISA for ACM

For nAb specificity tests, the recommended standard curves of FGF1 and FGF2 from their respective rat-specific ELISAs (Antibodies Online) were preincubated in 0, 10 or 20 µg/ml nAb at 37°C for 1 hr before being quantified in the ELISA according to manufacturer's instructions. ACM from CORT (n = 3) and EtOH (n = 3) treated astrocytes was analyzed using the same rat FGF2 ELISA (Antibodies Online, Atlanta, GA).

### Fear conditioning

Rats were immobilized for 3 hr or left undisturbed, then injected with 200 mg/kg BrdU. Either 2 days or 2 weeks after immobilization, rats were trained in contextual fear conditioning as per (**Kirby et al., 2012b**; 2 days: n = 10 con, n = 10 immob; 2 weeks: n = 20 con, n = 18 immob). The next day, rats were exposed to the fear conditioning context five times for 10 min each time with no shock (extinction). 24 hr later, they were re-exposed to the conditioning chamber for 10 min with no shock (extinction probe). A subset of rats (2 days: n = 10 con, n = 10 immob; 2 weeks: n = 10 con, n = 8 immob) was tested on an elevated plus maze 1 day prior to contextual fear conditioning (data not shown).

### Statistics

In most studies, data were analyzed using a one-way ANOVA followed by Dunnett's posthoc tests with the appropriate control group as reference. For in vivo CORT injection studies with 30 min and 3 hr time points, data were analyzed using two-way ANOVA followed by Dunnett's posthoc test (0 mg/kg-30 min used as reference). For cell culture studies, data were analyzed using two-way ANOVA followed by Dunnett's posthoc tests. For BLA lesion studies, a repeated measures two-way ANOVA was used with hemisphere being the paired variable. When only two groups were being compared, unpaired t-tests were used in all cases. $p \leq 0.05$ was considered significant.

## Acknowledgements

EDK was supported by a CIRM predoctoral fellowship and DoD NDSEG fellowship; LAB was supported by NSF GRFP.

# Additional information

## Funding

| Funder | Grant reference number | Author |
|---|---|---|
| National Institute of Mental Health, Biobehavioral Research Awards for Innovative New Scientists (NIMH BRAINS) | R01 MH087495 | Daniela Kaufer |
| National Alliance for Research on Schizophrenia and Depression (NARSAD) Young Investigator Award | R01 NS066005 | Daniela Kaufer |

The funders had no role in study design, data collection and interpretation, or the decision to submit the work for publication.

## Author contributions

EDK, Conception and design, Acquisition of data, Analysis and interpretation of data, Drafting or revising the article; SEM, Acquisition of data, Analysis and interpretation of data; WGS, Conception and design, Acquisition of data, Analysis and interpretation of data; DC, Conception and design, Acquisition of data, Drafting or revising the article; MJL, Acquisition of data, Drafting or revising the article; LAB, Acquisition of data, Drafting or revising the article; DK, Conception and design, Drafting or revising the article

## Ethics

Animal experimentation: All animal procedures were approved by the UC Berkeley Animal Care and Use Committees and performed in accordance with the recommendations in the Guide for the Care and Use of Laboratory Animals of the National Institutes of Health. All surgery was performed under sodium pentobarbital anesthesia with appropriate pre and post-operative analgesia, and every effort was made to minimize suffering.

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
