## [Decision Letter]

Thank you for choosing to send your work entitled “Acute stress enhances adult hippocampal neurogenesis and memory via secreted astrocytic FGF2 in rats” for consideration at *eLife*. Your article has been evaluated by a Senior editor and 3 reviewers, one of whom is a member of our Board of Reviewing Editors.

The Reviewing editor and the other reviewers discussed their comments before we reached this decision, and the Reviewing editor has assembled the following comments to help you prepare a revised submission. The reviewers agree that this is a thorough and interesting study, but they have the following suggestions:

1) This study reveals a new domain of the inverse-U phenomenon in stress and glucocorticoid action. To make the paper more broadly appealing, this should be discussed in the Introduction.

2) The Introduction should also include comments about the BLA given its role in mediating interactions between the amygala and hippocampus.

3) To be convinced on the causal role of astrocytic FGF2, additional data are needed to demonstrate that the in vivo increase in FGF2 is from astrocytes. Since neonatal astrocyte cultures are often contaminated, these sources need to be excluded using, e.g., in situ hybridization or immunohistochemistry on hippocampus tissue of similarly aged animals in vivo.

4) A related concern is the certainty that no other members of the FGF family are involved. Is the FGF2 blocking antibody specific to that form?

5) In addition, can you show actual levels of FGF2 present rather than relative levels, and is any FGF2 released from untreated astrocyte cultures (an important question since untreated ACM has no effect on NPC proliferation)?

6) The fear extinction data are interesting, but they feel like they belong in a different paper.

---

## [Author Response]

*1) This study reveals a new domain of the inverse-U phenomenon in stress and glucocorticoid action. To make the paper more broadly appealing, this should be discussed in the Introduction*.

We have added a paragraph introducing and discussing the inverted U function as the second paragraph of the Introduction. In addition, we have also discussed how this function potentially relates to previous literature on adult neurogenesis in the fifth paragraph and to our findings in the sixth paragraph.

*2) The Introduction should also include comments about the BLA given its role in mediating interactions between the amygala and hippocampus*.

Comments about the role of the BLA in influencing hippocampal stress responses and adult neurogenesis have been added to the third paragraph of the Introduction.

*3) To be convinced on the causal role of astrocytic FGF2, additional data are needed to demonstrate that the* in vivo *increase in FGF2 is from astrocytes. Since neonatal astrocyte cultures are often contaminated, these sources need to be excluded using, for example,* in situ *hybridization or immunohistochemistry on hippocampus tissue of similarly aged animals* in vivo.

We have added new data on FGF2 protein expression in GFAP+ astrocytes in adult male rats in a new Figure 7 (former Figure 7 is now Figure 8 and so on). We measured FGF2 expression in astrocytes using double immunohistochemical labeling for FGF2 protein and the astrocyte marker, glial fibrillary acidic protein (GFAP), in fixed hippocampal tissue sections from rats described in Figure 2 (con, immob, 0 mg/kg and 40 mg/kg CORT), all perfused 3 hr after immobilization or injection, as appropriate.

Consistent with previous reports [1], FGF2-immunoreactive cells were found throughout the DG and the hilus, with staining primarily in the cell body and nucleus. Z-stacks of the dentate gyrus and hilus were acquired using confocal microscopy, and used to identify GFAP+ vs GFAP- cells. The dentate gyrus is primarily composed of dentate granule neurons and while almost every cell expressed FGF2, we found very few GFAP+ cells within the DG, as expected. Given the ubiquitous and closely-packed FGF2 immunoreactivity in the DG, we had to measure mean optical density (with background correction) throughout the Z-stack of FGF2 expression in the DG (rather than in individual cells). No effect of stress or CORT injection on DG FGF2 expression was found using this measure.

In contrast, in the hilus, we could clearly identify and measure individual GFAP+ (and GFAP-) cells. Almost all GFAP+ cells were FGF2+. Given the mixed cell population but ubiquitous presence of FGF2 in GFAP+ cells, we measured integrated optical density of individual FGF2+ cells that were either GFAP+ or GFAP-. We found increased FGF2 signal in GFAP+ but not GFAP- cells in the hilus following both immobilization and 40 mg/kg CORT injection. These findings suggest that the FGF2 increase seen in whole hippocampal lysates by real time quantitative PCR and western blot (Figure 6) are most likely due to astrocytic FGF2 secretion. Moreover, while other cell types in other hippocampal subregions not quantified here could contribute to overall changes in FGF2, the DG and hilus are the regions flanking the neurogenic niche in the subgranular zone, and are therefore the most likely sources of growth factor changes that influence neurogenesis. These data are presented in the Results section following the section describing Figure 6. The relevant methods have also been added to the Materials and methods section where appropriate.

*4) A related concern is the certainty that no other members of the FGF family are involved. Is the FGF2 blocking antibody specific to that form*?

Of all the 23 members of the FGF family, FGF2 is the most potent growth factor for NPCs [2]. It is also one of the few FGFs known to be predominantly secreted by astrocytes (as opposed to neurons) [2] in the adult CNS. However, FGF1 has been reported to both stimulate NPCs [3] and be secreted from adult CNS astrocytes [4]. Previous work using the FGF2 nAb that we used in the present studies confirm that the nAb does not cross react with bovine or human FGF1 [5–7]. To confirm that the nAb does not neutralize rat FGF1, we performed ELISAs with either rat FGF2 or rat FGF1 and compared standard curves of protein dilutions with or without preincubation with the nAb. We found that while FGF2 was significantly neutralized at all concentrations tested, FGF1 availability in the assay was not significantly affected by the nAb. These data are now presented in Figure 9B,C. Given these specificity data and the prominent, nearly exclusive role of FGF2 in adult NPC proliferation, we feel relatively confident that the effects of this FGF2 nAb are specific to FGF2 neutralization in these studies.

*5) In addition, can you show actual levels of FGF2 present rather than relative levels, and is any FGF2 released from untreated astrocyte cultures (an important question since untreated ACM has no effect on NPC proliferation)*?

The FGF2 levels are now expressed in pg/ml from a rat FGF2 ELISA. Consistent with previous work, when astrocytes were only exposed to vehicle, FGF2 levels were similar to blank, suggesting very low basal secretion of FGF2 [8]. However, when treated with 1 μM CORT, FGF2 was found in the media at 3.5 pg/ml on average (Figure 9A). To confirm that FGF2 at concentrations in this range were sufficient to stimulate NPC proliferation, we treated adult rat NPCs with 0 and 4 pg/ml rat recombinant FGF2 for 3h, pulsed with BrdU and then fixed the cells 2h later (design in Figure 5A). We found that as little as 4 pg/ml rat rFGF2 increased the percent of BrdU+ NPCs (reported in the text of the results section for Figure 9). These data demonstrate that the levels of FGF2 found in ACM after CORT stimulation are sufficient to enhance NPC proliferation.

*6) The fear extinction data are interesting, but they feel like they belong in a different paper*.

Though the fear extinction data do represent a departure from the mechanistic data focusing on time points close to the stressor, we feel that including these data, and in particular the new neuron activation findings, are important to the behavioral relevance of the rest of the manuscript. Adult NPCs require 2–4 weeks to mature and become physiologically active in the adult rat [9]. Therefore, the relevance of an enhancement in proliferation immediately after stress is questionable without some longer-term effects. Since we do not see more new neurons persisting in immobilized rats two weeks after the stressor (Figure 10G,H), it might seem that the effect of acute stress does not have any functional output.

However, our finding that the neurons that have survived over the two weeks activate more in response to fear extinction in immob than control rats, and that these rats have a coincident behavioral memory benefit that does not appear at earlier time points, strongly suggests that the enhanced neurogenic potential of the hippocampus, following acute stress, has important behavioral relevance. Including behavioral and neuron activation data such as this is a common practice in the adult neurogenesis literature [10–12]. We therefore request to keep these data in the manuscript as we feel many researchers in the adult neurogenesis field will find them helpful in interpreting our data and integrating it into their understanding of how both stress and adult neurogenesis relate to memory and hippocampal function. We have added some additional explanation to the Figure 10 results sections to emphasize these points.

References

1. Bhatnagar, M et al. Neurochemical changes in the hippocampus of the brown norway rat during aging. Neurobiol Aging 18, 319–27 (1997)

2. Reuss, B. & von Bohlen un Halbach, O. Fibroblast growth factors and their receptors in the central nervous system. Cell and Tissue Research 313, 139–57 (2003) 10.1007/s00441-003-0756-7

3. Cheng, X et al. Acidic fibroblast growth factor delivered intranasally induces neurogenesis and angiogenesis in rats after ischemic stroke. Neurological Research 33, 675–80 (2011) 10.1179/1743132810Y.0000000004

4. Lee, M et al. Acidic fibroblast growth factor (FGF) potentiates glial-mediated neurotoxicity by activating FGFR2 IIIb protein. Journal of Biological Chemistry 286, 41230–45 (2011) 10.1074/jbc.M111.270470

5. Wang, L et al. A novel monoclonal antibody to fibroblast growth factor 2 effectively inhibits growth of hepatocellular carcinoma xenografts. Molecular Cancer Therapy 11, 864‐872 (2012) 10.1158/1535-7163.MCT-11-0813

6. Amano, O, Yoshitake, Y, Nishikawa, K & Iseki, S. Immunocytochemical localization of basic fibroblast growth factor in the rat pituitary gland. Archives of Histology and Cytology 56, 269–76 (1993)

7. Matsuzaki, K, Yoshitake, Y, Matuo, Y, Sasaki, H & Nishikawa, K. Monoclonal antibodies against heparin-binding growth factor II/basic fibroblast growth factor that block its biological activity: Invalidity of the antibodies for tumor angiogenesis. Proc Natl Acad Sci USA 86, 9911–5 (1989)

8. Forget, C, Stewart, J & Trudeau, L-E. Impact of basic FGF expression in astrocytes on dopamine neuron synaptic function and development. Eur J Neurosci 23, 603–16 (2006) 10.1111/j.1460-9568.2006.04570.x

9. Snyder, JS et al. Adult-born hippocampal neurons are more numerous, faster maturing, and more involved in behavior in rats than in mice. J Neurosci 29, 14484–95 (2009) 10.1523/JNEUROSCI.1768-09.2009

10. Epp, JR, Haack, AK & Galea, LA. Activation and survival of immature neurons in the dentate gyrus with spatial memory is dependent on time of exposure to spatial learning and age of cells at examination. Neurobiology of Learning and Memory 95, 316–25 (2011) 10.1016/j.nlm.2011.01.001

11. Farioli-Vecchioli, S et al. The timing of differentiation of adult hippocampal neurons is crucial for spatial memory. PLoS Biology 6, 2188–204 (2008) 10.1371/journal.pbio.0060246

12. Kee, N, Teixeira, CM, Wang, AH & Frankland, PW. Preferential incorporation of adult-generated granule cells into spatial memory networks in the dentate gyrus. Nat Neurosci 10, 355–62 (2007) 10.1038/nn1847